# Application of Artificial Intelligence Technology in Plant MicroRNA Research: Progress, Challenges, and Prospects

**DOI:** 10.3390/ijms262411854

**Published:** 2025-12-09

**Authors:** Ruilin Yang, Hanma Zhang

**Affiliations:** School of Life Sciences, Chongqing Normal University, Chongqing 401331, China; 13880234069@163.com

**Keywords:** plant microRNA, artificial intelligence, deep learning, target gene prediction, ceRNA network, graph neural network, explainable AI, multi-omics integration, crop improvement

## Abstract

Plant microRNAs (miRNAs) are endogenous non-coding RNAs (~20–24 nucleotides) that regulate gene expression post-transcriptionally, playing critical roles in plant growth, development, and stress responses. This review systematically examines AI applications in plant miRNA research, tracing evolution from traditional machine learning to deep learning architectures. Plant miRNAs exhibit distinctive features necessitating plant-specific computational approaches: nuclear-localized biogenesis, high target complementarity (>80%), and coding region targeting. These characteristics enable more accurate computational prediction and experimental validation than animal systems. Methodological advances have improved prediction accuracy from ~90% (early SVMs) to >99% (recent deep learning), though metrics reflect different evaluation contexts. We analyze applications across miRNA identification, target prediction with degradome validation, miRNA–lncRNA interactions, and ceRNA networks. Critical assessment reveals that degradome data capture mixed RNA fragments from multiple sources beyond miRNA cleavage, requiring stringent multi-evidence validation. Similarly, fundamental ambiguities in lncRNA definition compound prediction uncertainties. Major challenges include severe data imbalance (positive to negative ratios of 1:100 to 1:10,000), limited cross-species generalization, insufficient model interpretability, and experimental validation bottlenecks. Approximately 75% of plant miRNA families in miRBase v20 lack convincing evidence, underscoring the need for rigorous annotation standards. Future directions encompass multimodal deep learning, explainable AI, spatiotemporal graph neural networks, and ultimately AI-driven de novo miRNA design, though the latter requires substantial advances in both computation and high-throughput validation. This synthesis demonstrates that AI has become indispensable for plant miRNA research, providing essential support for crop improvement while acknowledging persistent challenges demanding continued innovation.

## 1. Introduction

### 1.1. Biological Significance of Plant miRNAs

Since their discovery in Arabidopsis in 2002, plant microRNAs have emerged as central regulators in molecular biology. These ~20–24 nucleotide non-coding RNAs recognize target mRNAs through sequence complementarity, guiding RNA-induced silencing complexes to cleave or repress targets, thereby precisely regulating gene expression [1]. Their functional importance extends across virtually all major plant processes, including organogenesis, developmental transitions, and hormone signaling [2,3]. The *miR156*-*SPL* module controls developmental stage transitions, while *miR172* regulates flowering time by targeting *AP2*-like transcription factors [4]. These discoveries have established miRNAs as important targets for molecular breeding and crop improvement [5,6].

### 1.2. Essential Differences Between Plant and Animal miRNA Systems

As shown in Figure 1, plant and animal miRNAs differ fundamentally in biogenesis pathways and target recognition, necessitating plant-specific computational tools [7,8,9]. Plant miRNA processing occurs entirely within the nucleus via DCL1, producing miRNA/miRNA* duplexes through two sequential cleavages [10]. In contrast, animal systems require Drosha-mediated nuclear processing followed by cytoplasmic Dicer cleavage [10].

More critically, target recognition mechanisms differ profoundly. Plant miRNAs exhibit near-perfect complementarity with targets (>80% identity), primarily binding coding regions and inducing mRNA cleavage [11,12]. Animal miRNAs achieve regulation through partial seed region matching (nucleotides 2–8), binding 3′UTRs, and causing translational repression [11,12]. These distinctions confer computational advantages for plant research. The high complementarity reduces search space, enabling algorithms to focus on near-perfect matches rather than complex partial pairing, naturally lowering false-positive rates [13]. The endonucleolytic cleavage mechanism produces definitive molecular signatures—5′ monophosphate-bearing fragments captured by degradome sequencing—providing direct experimental validation [14]. By contrast, animal miRNA regulation through translational repression requires technically challenging approaches like ribosome profiling [15]. This convergence of high complementarity, enabling accurate prediction with endonucleolytic cleavage and straightforward validation,, renders plant miRNA research particularly amenable to computational methods. Consequently, animal-focused computational methods cannot be directly applied to plants, necessitating plant-specific tools exploiting these advantageous characteristics [15].

### 1.3. Limitations of Traditional Research Methods

Plant miRNA research faces substantial methodological challenges. Early identification relied on time-intensive cloning and Northern blotting with limited throughput [16]. High-throughput sequencing improved discovery, but computational analysis remains difficult for complex genomes [17]. Early tools like miRanda and TargetFinder aided prediction but produced excessive false positives [18]. A comprehensive evaluation of eight tools revealed that only three could process wheat’s ~17 Gb genome, highlighting scalability limitations [19].

Annotation reliability poses critical challenges. Approximately 75% of plant miRNA families in miRBase v20 lacked convincing evidence, with only 9.4% achieving high-confidence status in v21 [7]. To address this, the 2018 community criteria established requirements: precise stem-loop excision (≥16 bp, MFE ~ −20 kcal/mol), miRNA/miRNA* detection with 2-nt 3′ overhangs, expression thresholds (RPM ≥ 3–10), DCL1-mediated processing, and independent validation [7,20]. Modern tools implement confidence classification systems.

For clarity on data types, sRNA-seq profiles mature miRNAs [17,21]; genome sequencing enables MIRNA loci prediction [22,23]; degradome/PARE validates targets via 5′ fragments [13,14]; and mRNA-seq identifies targets [9,18]. Tools categorize by input: structure-based (genome) [22,23,24,25], expression-informed (genome + sRNA-seq) [19,21], or genome-independent (sRNA-seq only) [26,27]. This review indicates each tool’s data requirements to guide method selection.

### 1.4. Necessity and Development History of Artificial Intelligence Technology

Artificial intelligence has transformed plant miRNA research by addressing traditional limitations. AI methods automatically learn discriminative features from raw data, eliminating manual feature engineering; integrating multi-dimensional data (sequence, structure, energy, conservation, expression) within unified frameworks; modeling complex nonlinear relationships in miRNA biogenesis; and efficiently processing massive high-throughput datasets enabling genome-scale analysis [28,29,30,31,32].

AI applications in plant miRNAs evolved through three phases: the foundational phase (2011–2015) established feasibility with tools like PlantMiRNAPred [22]; the deep learning transition (2016–2020) adopted CNNs and RNNs, significantly improving performance [33,34]; and the current era (2021-present) introduces Transformers and graph neural networks for increasingly complex problems [35,36,37]. The following sections systematically examine this evolution from traditional machine learning through deep learning innovations to current challenges and future prospects.

## 2. Traditional Machine Learning Methods

### 2.1. Support Vector Machines: The Foundation Tool for Plant miRNA Prediction

Support vector machines emerged as the foundational approach for computational plant miRNA prediction, leveraging their strong theoretical grounding in statistical learning theory. The core principle involves finding an optimal hyperplane in a high-dimensional feature space that separates samples of different classes with maximum margin. For plant miRNA problems characterized by limited numbers of known positive examples, SVMs offered important advantages over deep learning methods that typically demand substantially larger training datasets [38].

The development of PlantMiRNAPred in 2011 marked the entry of plant miRNA prediction into a specialized era, establishing several methodological principles that influenced subsequent tool development [22]. A pioneering observation established that plant and animal miRNA prediction methods fundamentally cannot be interchangeable, necessitating the construction of a dedicated prediction framework for plants. Their key innovation involved systematically extracting pseudo-pre-miRNAs from protein-coding sequences of Arabidopsis and soybean as negative training samples, filling a critical gap in the negative sample construction strategy. This approach better simulates real genomic background noise than using random sequences, because stem-loop structure-like fragments extracted from coding sequences represent naturally occurring “decoys” that computational methods must learn to discriminate from authentic miRNA precursors.

In terms of feature design, PlantMiRNAPred extracted multi-dimensional characteristics, including sequence composition captured through mononucleotide and dinucleotide frequency distributions, structural features encompassing stem-loop topology parameters and paired base ratios, and conservation features derived from cross-species comparisons. Through systematic feature selection procedures, the research identified the most discriminative feature subset optimized for classification performance. The model achieved accuracy exceeding 90% on eight plant species, including Arabidopsis, rice, poplar, *Physcomitrella patens*, alfalfa, sorghum, maize, and soybean, with sensitivity reaching 90.2% and specificity 95.1% [22]. More significantly, models trained on Arabidopsis data successfully predicted miRNAs of other plant species, demonstrating appreciable cross-species generalization capability. However, the method exhibited limitations, including complex feature engineering requirements, sensitivity to parameter choices, and the observation that cross-species applications often benefited from retraining with target species-specific data to achieve optimal performance.

The miPlantPreMat tool, developed in 2014, represented further innovation in SVM-based feature engineering [24]. This tool comprises two integrated modules: MiPlantPre for identifying pre-miRNAs from genome sequences and MiPlantMat for precisely locating mature miRNAs within identified precursors. The researchers proposed 152 novel sequence-structure-related features that systematically characterized multiple aspects of pre-miRNA properties. Through the application of the Back-SVM recursive feature elimination (B-SVM-RFE) method, the feature space was refined to 47 optimal features for pre-miRNA classification and 63 features for mature miRNA localization. This aggressive feature selection strategy not only improved prediction accuracy but also enhanced model interpretability, as researchers could gain biological insights from examining which sequence and structural properties emerged as most critical for miRNA recognition.

The tool’s negative sample construction strategy demonstrated particular ingenuity, extracting stem-loop structure-like fragments from protein-coding gene sequences as pseudo-pre-miRNA negative examples. This approach more faithfully simulates real background noise than random sequence generation, as these pseudo-hairpins exhibit structural features similar to authentic precursors while lacking true miRNA functionality. Testing on nine plant species yielded approximately 90% accuracy. A notable application to the tomato genome predicted 522 miRNAs, substantially exceeding the then-known 77 annotations, with subsequent experimental validation confirming many predictions [24]. Despite excellent performance metrics, miPlantPreMat retained limitations: the feature engineering process imposed substantial computational costs requiring calculation of RNA secondary structures and thermodynamic parameters; cross-species transfer performance required improvement; and recognition capability for miRNAs with atypical structures, such as multi-loop configurations, remained limited.

Continued systematic development in plant miRNA prediction produced the plantMirP series that represents a sustained refinement of SVM methodology. The plantMirP tool, published in 2016, introduced “knowledge-driven energy features” that marked a conceptual shift from purely statistical approaches [25]. Rather than relying solely on empirical feature extraction, the researchers designed five novel energy-related features grounded in an understanding of RNA folding thermodynamics. These features characterize essential properties of pre-miRNAs from a biophysical perspective, significantly improving discriminative power. Validation across multiple plant species demonstrated performance superior to contemporary tools, including miPlantPreMat and the original PlantMiRNAPred [25].

PlantMirP2, released in 2021, represents the current pinnacle of SVM methodology in plant miRNA prediction [23]. Key improvements included updated and expanded training datasets incorporating the latest miRBase annotations, further optimized knowledge-driven energy features reflecting a refined understanding of RNA thermodynamics, enhanced feature selection algorithms that better identified discriminative characteristics, and provision of more user-friendly web servers alongside standalone tools for different use cases. Performance metrics reached impressive levels: area under the receiver operating characteristic curve (AUC ~ area under curve) of 0.9968, accuracy of 97.54%, sensitivity of 96.75%, and specificity of 98.76%. The tool additionally integrated functionality for predicting miRNAs from next-generation sequencing data, expanding its applicability beyond genome-based prediction. The evolution of this tool series demonstrates how systematic integration of biological knowledge with machine learning methodology can progressively improve predictive performance [23].

Beyond general miRNA identification, researchers have developed SVM tools for specialized research applications. The microRPM tool exemplifies this approach, distinguished by its capability to predict miRNAs based solely on small RNA sequencing data without requiring reference genome sequences [26]. This represents a significant advance for miRNA research in non-model plants lacking genomic resources. The tool employs SVM to learn structural features characteristic of miRNA-miRNA* duplexes directly from sRNA-seq data, achieving 96.61% accuracy on Arabidopsis and 93.04% on rice. For experimental validation, researchers predicted 21 novel miRNAs in orchids and confirmed 18 through quantitative real-time PCR, yielding an 85.7% validation rate that substantiated the tool’s practical utility [26].

### 2.2. Random Forests and Ensemble Learning: Leveraging Multiple Perspectives

Random forest algorithms, implementing ensemble learning through the construction of multiple decision trees and integration of their collective predictions, have proven highly effective in plant miRNA research. This approach embodies the principle that combining multiple weak learners produces more stable and accurate predictions than any single model, analogous to the wisdom of the crowds phenomenon [39].

Precise localization of mature miRNAs within precursor sequences represents a critical aspect of miRNA functional research. The miRLocator tool, developed in 2015, with detailed usage protocols published in 2019, represents important progress in this domain [40,41]. miRLocator employs random forest algorithms while treating miRNA–miRNA* duplexes holistically rather than analyzing individual strands in isolation. The tool extracts position-specific features capturing nucleotide identity at each position, energy features quantifying thermodynamic stability, structural features describing secondary structure properties, and stability features assessing duplex robustness. This integrated approach recognizes that mature miRNA selection depends on the overall properties of the duplex structure rather than isolated sequence characteristics.

Testing across multiple species, including plants, demonstrated excellent performance, accurately predicting mature miRNA positions within precursor sequences. Subsequent work provided complete Python implementation code and a web interface, introducing automatic training functionality enabling users to train customized models using their own miRNA datasets [40,41]. This capability proves particularly valuable for candidate pre-miRNAs discovered from next-generation sequencing data, which may contain multiple potential mature miRNA positions. miRLocator efficiently narrows down candidates, substantially reducing subsequent experimental validation workload.

A comparative study of ensemble learning strategies systematically explored the performance of various ensemble methods in plant miRNA prediction [42]. The research compared four ensemble strategies—Bagging, Boosting, Stacking, and Blending—combined with five base classifiers, including Random Forest, Extra Trees, AdaBoost, Light Gradient Boosting Machine, and Support Vector Machine. On miRNA classification datasets, SVM achieved the highest accuracy of 0.659, while on pre-miRNA datasets, Extra Trees reached 0.67 accuracy, surpassing benchmark methods. A key finding revealed that ensemble strategies such as Stacking and Blending generally outperformed individual classifiers, and that algorithm selection significantly impacted performance [42]. These insights provide valuable guidance for researchers in selecting appropriate ensemble learning approaches for their specific applications.

For resource-limited non-model plants lacking reference genomes, the miRPreM method (2022) offers an innovative solution for resource-limited non-model plants [27]. Many economically important crops and wild plant species lack complete genome assemblies, rendering traditional prediction methods inapplicable. miRPreM adopts a bio-specific reference strategy, utilizing organism-specific or taxon-specific reference sequences rather than depending on full genome assemblies. This approach predicted 186 miRNAs compared to only 79 discovered by traditional methods, representing more than a twofold improvement. The methodology provides a feasible path for miRNA research in plant species with limited genomic resources, proving particularly valuable for medicinal plants and crop wild relatives [27].

### 2.3. Data Imbalance: A Pervasive Challenge and Its Solutions

In computational prediction of plant miRNAs, data imbalance represents a ubiquitous and serious problem that fundamentally affects classifier performance. This challenge manifests across multiple prediction tasks, including miRNA precursor identification, mature miRNA localization, and target gene prediction. The severity of the imbalance varies with the specific task: even in the model plant Arabidopsis, approximately 300 known MIRNA genes correspond to a limited number of pre-miRNA sequences. However, on a genome-wide scale, tens of millions of sequence fragments can fold into stem-loop-like structures that superficially resemble authentic miRNA precursors. This creates positive-to-negative sample ratios that may reach 1:10,000 or higher depending on the stringency of negative sample selection [43,44].

Severe data imbalance negatively impacts machine learning model training and performance through several mechanisms. Classifiers trained on imbalanced data tend to predict the majority class (negative samples) to maximize overall accuracy, resulting in severely insufficient recognition capability for the minority class (positive samples). On extremely imbalanced datasets, a simple rule-based classifier that predicts all samples as the negative class may achieve accuracy exceeding 99%, yet this inflated metric masks complete failure to identify any positive samples [45]. The model effectively learns to ignore minority class patterns, focusing instead on majority class characteristics.

A heterogeneous ensemble method published in 2013 represents an important breakthrough in addressing data imbalance for plant miRNA prediction [46]. Targeting pre-miRNA classification, this study proposed an innovative strategy combining improved SMOTE (Synthetic Minority Over-sampling Technique) with bagging approaches. The researchers introduced the concept of “self-containedness” and its derived features, which reflect the unique structural robustness of authentic pre-miRNAs. Genuine pre-miRNA stem regions tend to self-pair, forming stable structures that maintain this property even under perturbation, whereas pseudo-hairpins from random sequences typically exhibit less robust folding. This feature effectively distinguishes real pre-miRNAs from background pseudo-sequences.

The method’s core innovation lies in its heterogeneous ensemble strategy, combining three algorithmically distinct approaches: support vector machines, excelling at nonlinear boundary delineation, k-nearest neighbors, capturing local similarity patterns, and random forests, modeling feature interactions. More importantly, the researchers developed a modified-SMOTE bagging technique tailored to RNA sequence data characteristics, generating higher-quality synthetic samples than the standard SMOTE. The modified approach respects biological constraints of RNA structure, ensuring that synthetic samples maintain realistic secondary structure features and thermodynamic properties. Through synthetic oversampling of the minority class combined with multiple complementary base classifiers, the method achieved excellent performance: average accuracy of 96.54%, sensitivity of 94.8%, and specificity of 98.3% in rigorous cross-validation [46]. Notably, the method excelled in cross-species testing, achieving accuracy exceeding 93% for animal, plant, and viral pre-miRNAs. This research methodologically demonstrated the efficacy of heterogeneous ensembles for complex biological sequence classification problems, while innovations in feature engineering provided inspiration for subsequent investigations.

### 2.4. Systematic Comparison of Prediction Criteria and Weighting Schemes Across Tools

The fundamental question of what criteria different prediction tools employ and how these criteria are weighted profoundly impacts prediction reliability and interpretability. A systematic examination of representative tools reveals both shared principles and critical divergences that researchers must understand to appropriately select and interpret computational predictions.

The 2018 plant miRNA annotation criteria established by the community [7] provide a hierarchical framework that modern tools implement with varying emphasis. These criteria encompass structural requirements, including precise excision from stem-loop precursors with characteristic hairpin topology, typically requiring at least 16 paired nucleotides in the stem region and minimum free energy thresholds of approximately −20 kcal/mol [22,24]. Expression evidence is assessed through accumulation of both mature miRNA and miRNA* sequences exhibiting the characteristic 2-nucleotide 3′ overhang, with abundance thresholds varying from RPM (Reads Per Million) ≥ 3 to RPM ≥ 10 depending on desired confidence levels [47]. Processing signatures must demonstrate DCL1-mediated processing through precise 5′ and 3′ end determination [7], while independent validation requires detection across multiple datasets or experimental confirmation.

Different prediction tools assign dramatically different relative weights to these criteria, reflecting distinct philosophical approaches to balancing sensitivity versus specificity [48]. Sequence-structure-based tools like PlantMiRNAPred and plantMirP2 prioritize thermodynamic stability and structural features. Feature selection algorithms applied to these tools consistently reveal that MFE (Minimum Free Energy)-related features, including normalized MFE and MFE index, along with stem-loop topology parameters such as stem length and number of bulges, rank as top discriminative features collectively accounting for 40–60% of decision weight [22,23]. PlantMirP2’s knowledge-based energy features, designed based on an understanding of RNA folding thermodynamics, achieve 97.54% accuracy precisely because they capture the biophysical constraints that authentic miRNA precursors must satisfy [23].

Expression-based tools like miRDeep-P2 and mirador weigh read abundance and mapping precision most heavily. miRDeep-P2 implements a probabilistic scoring system where expression evidence contributes approximately 70% to the final score, while structural features contribute 30% [21]. miRador explicitly follows the 2018 annotation criteria as hard filters rather than weighted scores, implementing a sequential filtering pipeline where candidates must pass all structural checkpoints before expression thresholds are applied [48]. Hybrid approaches like miRScore attempt to balance multiple dimensions through quantitative thresholds: at least 75% precision in read alignment reflecting processing accuracy, enforcement of the 2-nucleotide 3′ overhang as a structural signature, and validation of expression levels above background noise [49].

Table 1 summarizes key quantitative thresholds employed by representative tools, illustrating the range of stringency levels applied in practice.

Systematic benchmarking demonstrates that no single weighting scheme dominates across all scenarios [19,48]. In a comprehensive evaluation, sRNAbench—which heavily weights expression patterns—achieved the highest precision but lower sensitivity, identifying fewer total miRNAs but with greater confidence in each prediction [19]. Conversely, miRDeep-P2—balancing structure and expression more evenly—demonstrated the highest sensitivity and processing speed, detecting more true miRNAs, including low-abundance species but with slightly elevated false-positive rates. The miRador tool, implementing strict adherence to 2018 community criteria, achieved precision comparable to other tools while reducing runtime by 2–10-fold-fold through efficient filtering logic [48]. The optimal tool choice depends critically on research goals: high-confidence annotation projects prioritize precision-optimized tools, while discovery-oriented studies benefit from sensitivity-optimized approaches that sacrifice some specificity to minimize false negatives.

The importance of appropriate criterion weighting becomes acute in cross-species applications. Targetfinder achieved 89% precision and 97% recall for Arabidopsis targets but only 46% recall for non-Arabidopsis species even after score optimization [50], indicating that Arabidopsis-trained weighting schemes fail to generalize to evolutionarily distant species. This performance degradation stems from species-specific variations in miRNA–target complementarity patterns, with non-Arabidopsis species exhibiting different distributions of mismatches in seed versus non-seed regions [50]. Tools must, therefore, either implement species-adaptive weighting through transfer learning approaches, use phylogeny-informed priors to adjust feature importance for taxonomically distant species, or adopt more conservative universal criteria that sacrifice some sensitivity for cross-species robustness.

Understanding these criterion–weight relationships enables evidence-based tool selection rather than arbitrary choices. Researchers should match tools to their data characteristics: expression-weighted tools excel with high-depth sequencing, while structure-weighted tools suit genome-only predictions. Species phylogeny matters critically, as tools trained on closely related species perform best, with performance degrading proportionally to phylogenetic distance. Annotation goals determine appropriate stringency: strict criterion enforcement for reference-quality annotations, relaxed thresholds for hypothesis generation. Computational resources must also be considered, as runtime-memory trade-offs vary 10–100-fold between tools. For critical applications, ensemble approaches combining complementary tools provide optimal results: the intersection of psRNATarget and Tapirhybrid delivers highly precise predictions, while the union of Targetfinder and psRNATarget maximizes sensitivity [50].

### 2.5. Practical Accessibility and Long-Term Sustainability of Traditional Machine Learning Tools

Beyond algorithmic sophistication, the practical utility and longevity of plant miRNA prediction tools depend critically on implementation quality, documentation completeness, and sustained maintenance commitment. A systematic assessment of representative traditional machine learning tools reveals substantial heterogeneity across these dimensions, with direct implications for their adoption and continued relevance in the research community.

PlantMiRNAPred, despite its pioneering role as the first plant-specific SVM predictor published in 2011, exemplifies the fragility of research software lacking institutional support. The tool has become unavailable over time, as explicitly noted in subsequent comparative studies that could only utilize its previously published results for benchmarking purposes rather than directly running the software [23]. This discontinuity represents a significant loss to the community, as PlantMiRNAPred’s carefully constructed negative sample strategy and systematic feature engineering established methodological principles that influenced subsequent tool development. The absence of code repositories or sustained web servers highlights a recurring challenge in bioinformatics: tools published in peer-reviewed journals may become inaccessible within years of publication when authors transition to different research areas or institutions discontinue server hosting.

In contrast, the plantMirP series demonstrates the benefits of sustained research group commitment and iterative tool refinement. The original plantMirP introduced knowledge-driven energy features and provided both command-line tools and moderate documentation, though installation complexity required substantial user expertise in dependency management [25]. The updated plantMirP2 addressed these usability concerns by offering dual interfaces—a user-friendly web server for exploratory analysis and standalone tools for high-throughput applications—alongside comprehensive documentation covering installation, parameter tuning, and worked examples [23]. The tool was released with both web server and standalone implementations, with source code made available through GitHub (https://github.com/wuqiansibai/plantMiRP2, accessed on 22 November 2025). Users should note that repository activity shows the last major updates in 2021, and web service availability may vary over time. Readers are advised to verify current tool accessibility through the GitHub repository or by consulting recent publications citing the tool. This dual-interface strategy was designed to balance accessibility for experimental biologists against flexibility for computational researchers.

miPlantPreMat occupies an intermediate position, with source code availability but limited long-term maintenance. While the tool’s sophisticated feature engineering—incorporating 152 sequence-structure features refined to 47 optimal descriptors through recursive feature elimination—demonstrated technical excellence, practical deployment faces obstacles. The GitHub repository provides code but lacks recent updates, installation instructions are incomplete with undocumented dependency version requirements, and no containerized distribution exists to ensure reproducibility across computing environments [24]. Consequently, although the tool theoretically remains accessible, successful deployment demands substantial troubleshooting expertise that limits practical adoption, particularly among users without extensive bioinformatics training.

microRPM represented an alternative deployment model optimized for resource-constrained non-model plant research. By implementing genome-independent prediction requiring only small RNA sequencing data, the tool addressed a critical need for species lacking reference genome assemblies [26]. However, the web server originally available at http://microRPM.itps.ncku.edu.tw is no longer accessible, and no alternative access method has been identified. This case illustrates the challenges of tool longevity in bioinformatics, where valuable methodologies may become unavailable within years of publication due to discontinued server hosting. However, web-only access introduces limitations: throughput constraints restrict large-scale genome-wide analyses, dependency on continuous institutional server hosting creates long-term availability uncertainty, and the absence of standalone software prevents integration into automated analysis pipelines. These trade-offs reflect broader tensions in tool design between maximizing accessibility and preserving analytical flexibility.

miRLocator exemplifies contemporary best practices in reproducible bioinformatics tool development. Beyond strong predictive performance, the tool distinguishes itself through comprehensive accessibility infrastructure: a Docker image (https://hub.docker.com/r/malab/mirlocator, accessed on 22 November 2025) encapsulates all dependencies in a portable container, ensuring identical execution environments across platforms, a web interface (http://miRLocator.omicstudio.cloud, accessed on 22 November 2025, users should verify current availability) provides immediate usability without installation, complete source code availability on GitHub (https://github.com/cma2015/miRLocator, accessed on 22 November 2025) enables algorithmic inspection and modification, and detailed protocol publications in Methods in Molecular Biology volumes provide step-by-step usage instructions [40,41], Note: A web interface was previously available at http://miRLocator.omicstudio.cloud but appears to be no longer operational as of November 2025; users should rely on the Docker image or source code installation instead. This multi-layered dissemination strategy addresses diverse user needs while prioritizing long-term reproducibility—a standard that remains aspirational for many bioinformatics tools. The containerization approach proves particularly valuable as it insulates users from dependency management challenges that frequently render older tools non-functional as underlying Python and R ecosystems evolve through breaking changes.

Random forest-based target prediction integrations, while algorithmically sophisticated, remain predominantly accessible through incorporation into larger frameworks rather than as standalone tools. The PAREameters tool provides an R package with comprehensive documentation and worked examples, facilitating data-driven inference of species-specific targeting rules from degradome data [51]. However, its command-line interface and R programming requirements present barriers for wet-lab biologists, and the lack of graphical user interface wrappers limits adoption outside bioinformatics-proficient groups. SmartPARE similarly provides valuable R-based functionality for degradome analysis but assumes users possess sufficient computational literacy to navigate installation procedures and troubleshoot common errors [52].

These implementation patterns reveal a fundamental tension in tool development between maximizing immediate accessibility and ensuring long-term utility. Web servers offer zero-barrier entry but create dependencies on sustained hosting commitments that frequently fail as funding cycles end. Command-line tools enable high-throughput automation and pipeline integration but demand technical expertise that excludes many potential users. Docker containers promise reproducibility but require infrastructure support and add computational overhead. Optimal dissemination likely requires hybrid strategies exemplified by plantMirP2 and miRLocator, providing multiple access modalities tailored to diverse user capabilities while prioritizing containerization and version-controlled code repositories to ensure multi-decade accessibility.

For practitioners selecting traditional machine learning tools, several decision criteria prove essential. First, verify active maintenance status through recent repository commits or responsive web server functionality rather than relying solely on publication date. Second, prioritize containerized distributions or web interfaces unless pipeline integration necessitates command-line tools. Third, assess documentation comprehensiveness, including troubleshooting guidance and expected outputs for test datasets. Fourth, ensure trained model parameters and preprocessing scripts are accessible, not merely algorithmic descriptions. For resource-limited groups, web servers like microRPM and plantMirP2 offer immediate productivity, while research labs developing analysis pipelines should prioritize actively maintained tools with container support like miRLocator to ensure long-term reproducibility as computational environments inevitably evolve.

## 3. Deep Learning Methods

### 3.1. Convolutional Neural Networks: Extracting Local Patterns with Global Understanding

Convolutional neural networks, originally designed for image analysis, possess architectural properties that translate remarkably well to RNA sequence analysis. Although RNA sequences represent one-dimensional data rather than two-dimensional images, local sequence motifs prove crucial for biological function, and these motifs may appear at variable positions along the sequence—characteristics analogous to local features in visual data that CNNs excel at detecting [53].

The development of mirDNN represents a significant breakthrough in applying deep learning to plant miRNA prediction [54]. This method, published in 2021, specifically addressed a problem that had long plagued genome-wide miRNA discovery: existing methods exhibited extremely high false-positive rates when applied to complete genomic sequences of new species. Although performance appeared acceptable on carefully curated test sets, precision fell far below reported values in actual genome-wide scanning applications.

mirDNN adopts a convolutional deep residual neural network architecture inspired by ResNet, the renowned computer vision model. The introduction of residual connections solved the gradient vanishing and degradation problems that typically emerge in deep networks, allowing the construction of deeper architectures with enhanced learning capacity. The model uses both RNA sequences and predicted secondary structure information as inputs, automatically learning discriminative features through multiple convolutional and residual blocks without requiring manual feature specification.

The architectural specifications of mirDNN illustrate the sophistication of modern deep learning approaches. The network accepts dual-channel input: sequence information encoded as a 4 × L matrix representing nucleotides A, C, G, and U, and structure information encoded as a 3 × L matrix representing left bracket (5′ paired), right bracket (3′ paired), and dot (unpaired) positions. Three convolutional blocks process this input sequentially, with filter numbers increasing from 64 to 128 to 256, and kernel sizes progressively expanding from four to five to six to capture increasingly complex patterns. Each block includes batch normalization and ReLU activation followed by max pooling, with residual connections bypassing blocks to maintain gradient flow. Fully connected layers with 512 and 256 neurons, each employing 50% dropout for regularization, integrate the extracted features before final classification.

Training employs focal loss to address class imbalance, with Adam optimization using a learning rate of 0.001. The researchers designed rigorous validation methods ensuring that reported performance reflects achievable results on truly novel species rather than overfitting to test sets. Testing on complete genome sequences of multiple animal and plant species, rather than merely on curated positive and negative example sets, revealed that at equivalent recall rates, mirDNN’s precision exceeded other methods by fivefold. The model achieved 87.02% accuracy across 11 species, including animals, plants, and viruses, demonstrating robust cross-kingdom capability [54].

To enhance interpretability—a persistent challenge for deep learning approaches—mirDNN provides nucleotide importance visualization functionality. By analyzing model attention weights, researchers can identify sequence regions most critical for prediction, yielding biological insights into miRNA recognition mechanisms. The tool has been open-sourced with comprehensive documentation and provides a web demonstration interface, facilitating adoption by researchers without deep learning expertise [54].

The CIRNN approach published in 2020 explores a synergistic combination of CNN and recurrent neural network architectures [55]. This research addressed the challenging task of plant miRNA–lncRNA interaction prediction, recognizing that plant non-coding RNA structures exhibit greater complexity than animal systems, rendering animal-focused models inapplicable. The limited data scale and high noise levels further constrained traditional machine learning approaches.

CIRNN’s innovation resides in its multi-level architectural design combining complementary network types. The CNN module automatically explores functional features of sequences, capturing local motifs and patterns through convolutional operations. The IndRNN (Independently Recurrent Neural Network) module obtains holistic sequence representations, learning dependencies between sequence positions while overcoming gradient vanishing problems that plague standard RNNs through architectural modifications, enabling neurons to process inputs independently. Through this combination, the model simultaneously leverages local motif information and global sequence dependencies. Experimental results demonstrated that CIRNN outperformed shallow machine learning methods, including SVM and Random Forest, as well as other deep learning models, on plant datasets, with advantages particularly pronounced for long sequence data characteristic of lncRNAs [55].

The miRDeep-P2 tool represents important progress in making deep learning methods practical for plant miRNA transcriptome analysis [21]. This tool specifically addresses plant-specific challenges through optimized filtering strategies and scoring algorithms that integrate the latest plant miRNA annotation standards. Particularly noteworthy, miRDeep-P2 effectively handles the complexity of large plant genomes, demonstrating high efficiency across species, including Arabidopsis, rice, tomato, maize, and wheat. The tool addresses a critical discrimination problem: many other types of small RNAs, particularly short interfering RNAs, are frequently misannotated as miRNAs in less sophisticated pipelines. Through improved filtering strategies informed by current annotation criteria, miRDeep-P2 significantly reduces false-positive rates and maintains computational efficiency even on the wheat genome, which at approximately 17 gigabases represents one of the most challenging plant genomic substrates [21].

### 3.2. Recurrent Neural Networks: Modeling Sequential Dependencies

Recurrent neural networks are specifically architected for processing sequential data, maintaining internal states that encode information from previous inputs and thereby capture temporal or positional dependencies. In plant miRNA research, RNA sequences are inherently ordered, with nucleotide arrangement critically determining biological function, making RNNs a natural architectural choice [56].

The DIGITAL tool (2023) represents a successful application of Long Short-Term Memory networks—a sophisticated RNN variant—to the plant-specific problem of identifying miRNA-triggered phasiRNA (phased siRNA) loci [57]. PhasiRNAs constitute a special class of plant small RNAs produced through miRNA-triggered cleavage of PHAS genes, playing important roles in plant reproductive development and disease resistance. Identifying PHAS loci proves crucial for understanding phasiRNA biogenesis and function, yet this represented an unaddressed gap before the first specialized deep learning tool emerged.

DIGITAL employs LSTM architecture to capture long-distance dependencies within sequences, learning sequence features characteristic of *PHAS* loci without requiring manual feature specification. The tool uses one-hot encoding to convert nucleotide sequences into numerical representations suitable for neural network processing, achieving end-to-end learning with direct prediction from raw sequences. On two independent test sets differentiated by sequence length, DIGITAL achieved accuracies of 98.48% and 94.02%, significantly outperforming traditional machine learning baselines. The tool demonstrates robustness and strong generalization ability, proving applicable to species beyond its training set, and has been open-sourced to facilitate use and extension by the research community [57].

The PmliPEMG method published in 2021 represents a sophisticated application of deep learning ensemble approaches to plant ncRNA interaction prediction [58]. Addressing the lack of reliable cross-species tools for plant miRNA–lncRNA interaction prediction, this research proposed a multi-level information enhancement and greedy fuzzy decision ensemble model. PmliPEMG adopts a CNN-LSTM hybrid network architecture where multi-scale convolutional layers extract local features of varying lengths, LSTM layers model long-distance dependencies across sequences, and attention mechanisms adaptively weight important features.

The innovation resides in the multi-level information enhancement framework operating at three levels. At the feature level, the model fuses complex features spanning sequence composition, predicted secondary structure, and thermodynamic parameters. At the scale level, multi-scale convolution captures patterns of different granularities simultaneously. At the model level, attention mechanisms focus computational resources on key informative regions. Additionally, researchers proposed a novel greedy fuzzy decision ensemble strategy that improves computational efficiency while maintaining performance by adaptively selecting complementary base learners. Experimental results demonstrated that the model outperformed existing state-of-the-art prediction methods and exhibited strong cross-species generalization capability [58].

### 3.3. Transformer Architecture: Leveraging Attention Mechanisms for Long-Range Interactions

The Transformer architecture has achieved revolutionary success in natural language processing in recent years, with its self-attention mechanism directly modeling relationships between any two positions in a sequence. This capability proves particularly suitable for capturing long-distance base pairing interactions fundamental to RNA structure and function [59].

Traditional RNNs process sequences sequentially, requiring information to propagate through many computational steps to model long-range dependencies. This sequential processing creates two problems: computational bottlenecks preventing parallelization and gradient degradation over long distances. Transformers address both limitations through self-attention, which computes relevance scores between all position pairs simultaneously. For RNA sequences, this proves invaluable because complementary pairing can occur between nucleotides separated by hundreds of positions, as in the stem regions of long precursor structures.

The PmlIPM tool (2023) introduced Transformers to plant miRNA–lncRNA interaction prediction [60]. The tool implements a four-component framework: input embedding, converting sequences into high-dimensional representations; positional encoding, preserving sequence order information; multi-head attention capturing, using diverse types of complementarity patterns; and max pooling, aggregating the most relevant features. The architectural ingenuity lies in the separate input strategy: PmlIPM independently receives miRNA and lncRNA inputs to extract sequence features, avoiding information loss that might result from directly concatenating sequences as model input.

The multi-head attention mechanism, typically employing 4–8 attention heads, endows the model with the capability to capture long-distance features through multiple parallel attention computations. Different attention heads can specialize in different aspects of the interaction: one head might focus on seed region complementarity, another on structural accessibility, a third on non-canonical pairing. This parallel processing of complementary information sources enables the model to understand potentially complex interaction patterns between miRNAs and long non-coding RNAs. Validation on benchmark datasets from four plant species—*Arabidopsis lyrata*, tomato, *Brachypodium distachyon*, and potato—yielded AUC scores of 0.8412, 0.8587, 0.9666, and 0.9225, respectively, demonstrating performance superior to existing methods [60]. This confirms the advantage of Transformers in handling complex tasks involving plant non-coding RNA interactions.

The PmiProPred tool (2025) combines Transformers with CNN for plant miRNA promoter prediction [61]. This research addresses transcriptional regulation of miRNA genes, which proves crucial for understanding miRNA expression control. The tool synergistically combines CNN and Transformer architectures, leveraging the Transformer’s global receptive field alongside CNN’s local feature extraction capability, further integrating Convolutional Block Attention Mechanism to enhance feature discrimination. The model achieved 92–97% accuracy across three independent test sets, representing a successful application of Transformer architectures to diverse aspects of plant miRNA research beyond interaction prediction [61].

### 3.4. Graph Neural Networks: Analyzing Network Topology

Graph neural networks represent deep learning architectures specifically designed for processing graph-structured data, offering natural solutions for problems inherently involving relationships and connections. RNA molecules can be naturally represented as graphs: nodes represent nucleotides, edges represent connections between nucleotides, including phosphodiester backbone bonds and hydrogen bond base pairing, and the graph structure directly encodes RNA secondary structure topology [62].

The CFHAN (Counterfactual Heterogeneous Attention Network), published in 2024, represents a cutting-edge GNN application in plant lncRNA–miRNA interaction prediction [63]. This research addressed low signal-to-noise ratios and robustness problems faced by existing GNN methods through innovative architectural design. The researchers constructed a heterogeneous network reflecting real biological systems, incorporating two different node types representing lncRNAs and miRNAs, along with interaction edges connecting them based on experimental evidence or computational predictions.

CFHAN’s core innovation implements three levels of attention mechanisms combined with counterfactual link enhancement. Node-level attention focuses on important features of individual nodes, weighting molecular properties like expression levels and conservation. Semantic-level attention identifies the importance of different relationship types in the heterogeneous network, recognizing that some interaction types carry more biological significance than others. Counterfactual link enhancement improves node embedding learning by introducing “what if” reasoning: considering how predictions would change if certain edges were modified provides robustness to noisy or missing interactions. These learned embeddings are subsequently input to a multi-layer perceptron for final interaction prediction.

Evaluation on benchmark datasets of plant lncRNA–miRNA interactions demonstrated that CFHAN surpassed five state-of-the-art methods, achieving excellent performance with an average AUC of 0.9953 and average accuracy of 0.9733. More significantly, the model exhibited strong cross-species prediction capability, providing value for non-model plants that lack abundant experimental interaction data [63]. This capability addresses one of the most pressing challenges in plant genomics: extending insights from well-studied model organisms to agriculturally important species with limited experimental characterization.

The MTAGCN tool (2022) applied graph convolutional networks to tea plant (Camellia sinensis) miRNA–target association prediction [64]. This research constructs miRNAs, target genes, and their interactions as a bipartite graph, performing convolution operations on this graph structure to aggregate neighborhood information. The introduction of layer attention mechanisms enables adaptive weight assignment to features from different network layers, enhancing model expressiveness. The tool represents the first GNN application to tea plant miRNA research, filling a methodological gap for this economically important species and providing new computational capabilities for miRNA functional research in non-model crops [64].

Although GNN applications in plant miRNA research remain relatively limited compared to CNN and RNN approaches, these preliminary explorations have already demonstrated unique advantages. RNA secondary and tertiary structures, miRNA regulatory networks, and ceRNA networks can all be naturally represented as graphs, making GNN an ideal choice for processing these inherently relational data types. With continued data accumulation and methodological maturation, GNNs are expected to play increasingly important roles in modeling miRNA family evolutionary relationships through phylogenetic networks, predicting miRNA functions based on network topology, and conducting systems biology analyses integrating multi-omics data.

Having examined individual AI architectures and their specific applications, Figure 2 synthesizes these methods into a comprehensive computational pipeline for plant miRNA research. The pipeline integrates diverse input data sources—small RNA sequencing for mature miRNA profiling, RNA-seq for transcriptome analysis, degradome sequencing for cleavage validation, and genome data for loci prediction—processing them through four specialized AI modules: miRNA identification using SVM/CNN approaches (plantMirP2, mirDNN), target prediction combining machine learning with degradome evidence (psRNATarget, PAREameters), interaction prediction via Transformer/GNN architectures (PmlIPM, MTAGCN), and network construction through graph analysis techniques (PceRBase, CFHAN). This integrated framework generates actionable outputs, including candidate miRNA sequences, curated target gene lists, reconstructed regulatory networks, and functional annotations, illustrating how AI has become indispensable across all stages of miRNA functional research.

### 3.5. Systematic Comparison of Deep Learning Architectures

While deep learning methods demonstrate superior performance over traditional machine learning, their practical application requires understanding the technical specifications that determine their behavior and computational requirements. A systematic examination of input representations, architectural parameters, and learning strategies across representative tools reveals both design principles and implementation trade-offs that guide rational tool selection for specific research objectives.

Different encoding schemes transform biological sequences into numerical representations, capturing distinct aspects of miRNA biology [65]. One-hot encoding represents each nucleotide as a binary vector—A as (1,0,0,0), C as (0,1,0,0), G as (0,0,1,0), U as (0,0,0,1)—providing the simplest raw sequence representation used by most CNN-based tools [54]. However, this approach discards structural information that may prove critical for recognition. Sequence-structure hybrid encoding addresses this limitation by combining nucleotide identity with predicted secondary structure symbols, creating multi-channel inputs where one channel encodes sequence, and another encodes structure, including paired bases and unpaired regions [65,66]. mirDNN exemplifies this approach, using both sequence and structure as parallel input channels to its convolutional layers [54].

K-mer frequency encoding captures local sequence patterns by counting occurrences of subsequences of length k, typically ranging from three to six nucleotides, transforming variable-length sequences into fixed-dimensional feature vectors. The kmerPMTF tool employs self-supervised deep neural networks to learn semantic embeddings from k-mer frequencies, enabling the capture of latent sequence patterns beyond simple counting statistics [67]. Multi-level feature hierarchies, as implemented in DeepMirTar for animal miRNA research, integrate three representation levels: high-level expert-designed features, including seed match patterns and free energy, low-level expert-designed features, encompassing site conservation and accessibility, and raw one-hot encoded sequences, totaling 750 features spanning abstract biological knowledge to raw data [68]. This hierarchical approach attempts to combine the interpretability of expert features with the pattern discovery potential of raw sequence learning.

Table 2 compares 5 deep learning architectures (CNN, CNN-RNN, LSTM, Transformer, GNN) for miRNA prediction, covering their tools, key parameters, inputs, training resources (time, GPU memory, RAM), and accuracy. Training used RTX 3000 GPUs (batch size 32–128, ~5–10 K samples); “GPU memory” refers to training VRAM (30% for inference). Resource needs vary by architecture (e.g., Transformer needs ≥16 GB GPU memory; GNN memory scales with graph size), and accuracy ranges reflect performance differences.

Network architecture specifications vary substantially across tools, reflecting different assumptions about miRNA sequence pattern properties [65]. For CNN-based approaches, convolutional layer depth typically ranges from 2 to 4 layers. mirDNN implements a deep residual architecture inspired by computer vision success stories, overcoming gradient vanishing in deeper networks through skip connections [54]. Kernel sizes follow biological intuition: initial layers use small kernels of size 2–4 to capture dinucleotide motifs, while deeper layers employ larger kernels of size 5–8 to detect longer structural patterns. mirDNN uses progressively increasing kernel sizes of four, five, and six across its three convolutional blocks [54]. Filter numbers increase with depth following patterns like 16→32→64 or 64→128→256, enabling detection of increasingly complex composite features [65,66]. Pooling strategies determine spatial invariance, with max pooling dominating due to its ability to capture the strongest signal regardless of position—essential for detecting miRNA motifs that may appear anywhere in the sequence [54].

For RNN-based tools, hidden layer dimensionality typically ranges from 64 to 256 units, balancing expressiveness against overfitting risk. DIGITAL uses LSTM with 128 hidden units for capturing long-range dependencies in phasiRNA loci sequences [57], while CIRNN combines CNN for local pattern extraction with IndRNN employing 64-unit layers for sequence representation learning. The IndRNN architecture overcomes traditional RNN gradient problems through neuron independence, allowing deeper stacking without vanishing gradients [55]. Transformer-based architectures like PmlIPM implement multi-head attention with 4–8 heads, each capturing different aspects of miRNA-lncRNA complementarity patterns, with embedding dimensions of 128–256 providing sufficient representational capacity [60].

Learning strategies and optimization approaches profoundly influence model performance and generalizability [69]. Standard supervised learning dominates miRNA identification tasks, using labeled miRNA and non-miRNA pairs with cross-entropy loss optimized via Adam or stochastic gradient descent with learning rates of 0.001–0.01 [54,57]. However, severe class imbalance with positive-to-negative ratios from 1:100 to 1:10,000 necessitates specialized strategies. Focal loss down-weights easy negatives to focus learning on hard examples, class-weighted loss assigns higher penalties to minority class misclassifications, and SMOTE-based oversampling synthetically generates minority class examples as developed in the Lertampaiporn heterogeneous ensemble study [46].

Self-supervised pre-training, employed by kmerPMTF, first learns general sequence representations from unlabeled data through reconstruction or contrastive objectives, and then fine-tunes on labeled miRNA data. This two-stage approach proves particularly valuable for low-resource species where labeled data are scarce [67]. Ensemble learning combines multiple models to reduce variance and improve robustness: PmliPEMG implements a greedy fuzzy decision ensemble that adaptively selects complementary base learners, achieving better performance than individual models [58]. Regularization techniques prevent overfitting through dropout rates of 0.3–0.7 randomly deactivating neurons during training, L2 weight decay penalizing large weights, and early stopping halting training when validation performance plateaus [54,65]. Batch sizes typically range from 32 to 128, balancing gradient estimate quality against memory constraints and training speed [66].

### 3.6. Systematic Advantages and Persistent Challenges of Deep Learning

Deep learning methods offer fundamental advantages over traditional machine learning that extend beyond raw performance metrics, representing a paradigm shift in how computational biology approaches sequence analysis (Figure 3). The capacity for automatic feature learning eliminates the need for labor-intensive manual feature engineering that dominated traditional approaches. Rather than requiring expert specification of which sequence patterns matter, deep networks discover discriminative representations directly from data. This proves particularly valuable when moving to new species or new biological questions, as the model can adapt its learned features rather than requiring complete re-engineering of the feature extraction pipeline.

The scalability advantages become apparent when processing large and complex genomes. Traditional SVM methods exhibit computational complexity ranging from O(n^2^) to O(n^3^) with respect to training set size, making genome-wide applications on species like wheat computationally prohibitive. Deep learning methods, particularly when trained with mini-batch stochastic gradient descent, achieve more favorable scaling properties. For instance, while only three out of eight traditional tools could process the wheat genome in the Li systematic evaluation [19], mirDNN successfully completed genome-wide scanning in reasonable timeframes [54].

Transfer learning capability represents another crucial advantage. Models pre-trained on data-rich species can be fine-tuned on target species with limited data, reducing required training samples by 60–80% according to various plant classification studies [70]. This proves invaluable for non-model organisms lacking extensive training data. Furthermore, multimodal integration—combining sequence, structure, expression, and other data types—proceeds more naturally in deep learning frameworks through end-to-end architectures that learn optimal feature fusion, whereas traditional methods require manual feature concatenation with ad hoc weighting schemes.

However, interpretability trade-offs remain significant [71,72]. Although models may achieve excellent prediction performance, their decision-making processes often resist straightforward interpretation, creating tension with biology research’s need for mechanistic understanding. Biologists require not merely accurate predictions but insight into why particular sequences are classified as miRNAs, which features prove most critical for miRNA biogenesis and function, and what testable biological hypotheses emerge from model behavior. While attention visualization approaches like those implemented in mirDNN provide partial solutions [54] and techniques like SHAP (SHapley Additive exPlanations) values offer post-hoc explanations, the fundamental challenge persists that deep learning’s power derives partly from its ability to learn representations that humans might not naturally conceive or easily interpret [73,74,75].

### 3.7. Critical Assessment of Deep Learning Tool Ecosystem and Reproducibility

The transition from traditional machine learning to deep learning methods in plant miRNA research promised unprecedented predictive accuracy through automatic feature learning and complex pattern recognition. However, this algorithmic sophistication introduces distinct practical challenges concerning computational resource requirements, code availability, reproducibility barriers, and long-term maintainability that deserve systematic examination.

mirDNN represents a notable exception to reproducibility challenges, providing multiple access pathways that facilitate adoption [54]. The method offers a web demonstration server (http://sinc.unl.edu.ar/web-demo/mirdnn/, accessed on 22 November 2025; users should verify current availability as web services may experience occasional downtime), enabling immediate testing without local installation, complete source code and trained models available through both GitHub (https://github.com/cyones/mirDNN, accessed on 22 November 2025) and SourceForge (https://sourceforge.net/projects/sourcesinc/files/mirdnn/, accessed on 22 November 2025) repositories, and detailed training and prediction protocols documented with example datasets. The web interface incorporates nucleotide importance visualization functionality, translating abstract model predictions into interpretable sequence-level insights that facilitate biological hypothesis generation. Critically, the tool explicitly specifies GPU requirements and provides CPU-compatible alternatives, acknowledging that not all research groups possess access to specialized hardware. The documentation candidly notes that training a model is computationally intensive and, therefore, recommends GPU usage, exemplifying transparent communication of resource requirements [54].

However, substantial computational demands for model training—requiring CUDA-compatible GPUs and multi-hour runtimes even on modern hardware—restrict de novo model development to well-resourced laboratories. Most users must rely on pre-trained models that may not generalize optimally to their specific plant species or genomic contexts. The challenge extends beyond mere hardware availability: configuring appropriate software environments with compatible versions of CUDA, cuDNN, TensorFlow, or PyTorch, and Python frequently demands substantial technical expertise.

In stark contrast, CIRNN exemplifies reproducibility failures affecting numerous deep learning publications. Despite reporting superior performance over existing methods in plant miRNA–lncRNA interaction prediction and publication in a peer-reviewed journal, no publicly accessible code repository exists for this CNN-IndRNN ensemble approach [55]. Exhaustive searches across GitHub, institutional repositories, and supplementary materials yield no implementation, effectively rendering the method inaccessible regardless of its algorithmic merits. A comprehensive survey of lncRNA–miRNA interaction prediction methods explicitly documents this availability gap, listing CIRNN with a downloadable publication PDF but no corresponding code availability [76]. This pattern—high-quality algorithmic innovation described in peer-reviewed publications yet lacking accessible implementations—fundamentally undermines scientific reproducibility and prevents community validation of reported performance claims.

DIGITAL demonstrates intermediate accessibility, with source code publicly available through GitHub, enabling algorithmic inspection and potential reuse [57]. However, the repository provides minimal documentation beyond basic installation instructions, lacks worked examples with expected outputs, and does not specify dependency versions precisely enough to ensure environment reproducibility. Users report challenges in recreating the exact computational environment necessary for successful execution, as undocumented version incompatibilities between TensorFlow, Keras, and CUDA libraries can silently alter results or cause cryptic runtime errors. The tool targets a specialized prediction task with more limited community demand than general miRNA prediction, potentially explaining reduced investment in user support infrastructure.

PmlIPM occupies an ambiguous position where code status remains unclear [60]. Repository searches yield no definitively associated GitHub presence, though the possibility exists that code is available under alternative naming conventions or hosted on institutional servers not indexed by standard search engines. This ambiguity itself constitutes a usability problem: researchers should not need to conduct detective work to locate implementations of published methods. The high GPU requirements noted in the publication—demanding modern NVIDIA GPUs with substantial VRAM—further constrain accessibility even if code eventually surfaces.

CFHAN, despite achieving exceptional reported performance with an AUC of 0.9953 in 2024, similarly lacks a publicly accessible implementation [63]. No GitHub repository, Docker container, or web server exists to enable independent validation or application of this graph neural network approach. The sophisticated architecture incorporating node-level, semantic-level, and counterfactual attention mechanisms likely demands substantial expertise to reimplement from publication descriptions alone, effectively placing the method beyond the reach of laboratories without specialized deep learning and graph neural network proficiency.

Computational resource requirements emerge as a systemic barrier limiting deep learning method adoption. Unlike traditional machine learning tools that typically execute on standard desktop workstations, state-of-the-art deep learning approaches routinely demand modern GPU acceleration with NVIDIA GPUs supporting CUDA, often requiring at least 8 GB VRAM for Transformer models and 16 GB for large graph neural networks. Substantial memory capacity of 32 GB or more system RAM proves necessary for preprocessing large genomic datasets. An extended runtime spanning hours to days for model training, even on GPU-accelerated hardware, compared to minutes to hours for traditional ML methods, imposes practical constraints. Specialized software stacks requiring specific version combinations of CUDA, cuDNN, TensorFlow, or PyTorch, and Python frequently exhibit breaking incompatibilities across updates, creating deployment challenges.

These requirements create a two-tier accessibility landscape where well-funded research institutions with dedicated computational infrastructure can leverage deep learning methods, while resource-limited groups—including researchers in many plant biology departments and institutions in developing countries—remain effectively excluded. Web-based deployments partially address this disparity by providing GPU-accelerated predictions without requiring local infrastructure, yet throughput limitations prevent large-scale genome-wide analyses that constitute common use cases.

Reproducibility challenges extend beyond code availability to encompass model weight portability, dependency specification, and platform compatibility. Deep learning models trained on specific GPU architectures may exhibit different numerical behavior or fail entirely when transferred to alternative hardware due to non-deterministic operations in GPU libraries, subtle differences in floating-point arithmetic across platforms, and framework-specific implementation details. Few tools provide model checkpoints—trained network weights—alongside source code, forcing users to retrain from scratch and potentially failing to reproduce originally reported performance if training data preprocessing steps were inadequately documented.

The plant miRNA community would benefit from mandating that all published deep learning methods include permanent code repositories with DOI assignment via Zenodo or figshare, pre-trained model weights in standardized formats, Docker containers specifying exact dependency versions, benchmark datasets with expected outputs enabling validation of correct installation, computational resource documentation, including GPU model and memory requirements with typical runtimes, and troubleshooting guides addressing common installation failure modes. Practitioners navigating the current landscape should prioritize tools with verified code availability, assess computational resource compatibility, consider hybrid approaches combining traditional ML for candidate identification with deep learning refinement where implementations exist, engage the research community by contacting tool authors when code availability is ambiguous, and plan for potential reimplementation if critical applications demand methods with inaccessible code.

**Figure 3 ijms-26-11854-f003:**
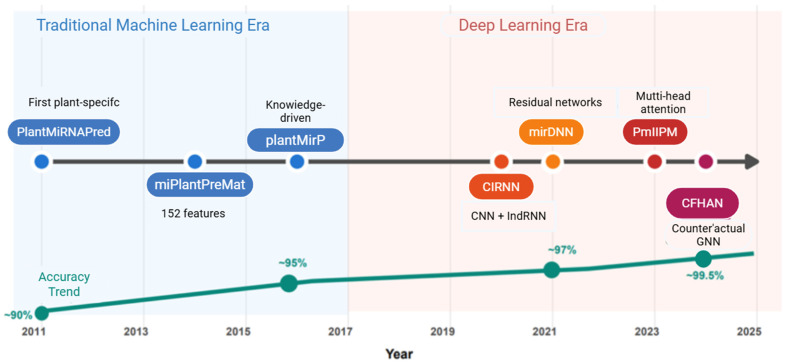
Evolution of AI Methods in Plant miRNA Research. This timeline visualization traces the development of artificial intelligence approaches for plant miRNA identification and analysis from 2011 to 2024, demonstrating the transition from traditional machine learning to deep learning architectures. The upper section delineates two distinct eras: the Traditional Machine Learning Era (2011–2016, light blue background) dominated by support vector machine (SVM)-based methods, including PlantMiRNAPred (2011), miPlantPreMat (2014), and plantMirP (2016), and the Deep Learning Era (2016–2024, light orange background) characterized by neural network approaches such as mirDNN (2021) utilizing convolutional neural networks with residual connections, CIRNN (2020) combining CNNs with independently recurrent neural networks, PmlIPM (2023) implementing transformer-based multi-head attention mechanisms, and CFHAN (2024) employing counterfactual graph neural networks. The lower curve tracks the concurrent improvement in prediction accuracy, rising from approximately 90% in 2011 to 95% in 2016, 97% in 2021, and reaching 99.5% in 2024. Color-coded markers distinguish different algorithmic categories (SVM in blue, CNN in orange, RNN in red, Transformer in dark red, and GNN in purple), with associated technical innovations noted for each milestone, illustrating the progressive shift from feature engineering-dependent methods to end-to-end deep learning systems capable of automatic representation learning.

## 4. Application Scenarios: From Identification to Networks

### 4.1. Evolution of Target Gene Prediction Methods

MicroRNAs regulate gene expression through complementary pairing with target gene mRNAs, making accurate prediction of miRNA–target relationships a central objective for understanding their biological functions. Methodologies in this field have evolved from simple sequence complementarity rules through machine learning enhancement to contemporary deep learning exploration, with each phase building upon insights from predecessors while addressing their limitations [77].

Early plant miRNA target gene prediction relied primarily on sequence complementarity rules, exploiting the distinctive high complementarity between plant miRNAs and their targets. Unlike animal miRNAs that achieve regulation through partial seed region matching, plant miRNAs typically exhibit greater than 80% complementarity with target genes, with binding sites predominantly localized to coding regions rather than 3′ untranslated regions. Tools, such as miRanda and TargetFinder, screened candidate targets through sequence alignment combined with free energy calculations, while psRNATarget integrated multiple information types, including sequence complementarity, minimum free energy, and evolutionary conservation, to become one of the most widely used online prediction platforms in plant biology [78,79]. However, approaches based solely on sequence complementarity characteristics produce substantial numbers of false-positive predictions, because many sequences exhibiting partial complementarity to miRNAs occur randomly throughout genomes without genuine regulatory relationships [80].

The introduction of machine learning methods significantly improved prediction accuracy by learning to distinguish functional targets from spurious sequence matches. A study published in 2014 represents an important milestone in this transition from rule-based to learning-based prediction [81]. This research proposed a principal component analysis-SVM ensemble method, using results from three online miRNA target prediction tools—miRanda, TargetFinder, and psRNATarget—as initial features. Principal component analysis performs dimensionality reduction and decorrelation of high-dimensional features, extracting principal components that capture the most discriminative information. Subsequently, SVM classifiers trained on these PCA-extracted features distinguish true targets from false positives. The study also adopted self-training techniques, using semi-supervised learning ideas to gradually expand the training set by iteratively adding high-confidence predictions.

The innovation of this method lies in its multi-tool integration strategy, recognizing that different prediction tools employ different algorithms and parameters, each with distinct strengths and weaknesses, such that integrating their results enables complementary information to improve overall accuracy. More importantly, researchers utilized degradome sequencing data as experimental validation evidence, providing high-confidence ground truth for supervised learning.

However, interpreting degradome data demand careful consideration of their inherent complexities and limitations. Degradome libraries, constructed by capturing polyadenylated RNA fragments with 5′ monophosphate termini, do not exclusively represent miRNA-directed cleavage products [13,14]. Rather, they capture a heterogeneous mixture of RNA ends generated through multiple cellular mechanisms. Systematic analysis revealed that degradome datasets contain RNA ends from diverse sources beyond miRNA-mediated cleavage [82]. These include processing intermediates from ribosomal RNA and small nucleolar RNA maturation pathways, cleavage fragments generated by RNA-binding proteins at their specific recognition motifs, products of general RNA decay pathways, and technical artifacts introduced during library construction, such as spurious 5′ ends from non-specific PCR amplification [82].

Importantly, the proportion of reads representing authentic miRNA–target cleavage versus other sources varies with sequencing depth and the specific transcript being examined. For high-abundance mRNAs, deeper sequencing progressively reveals lower-abundance cleavage events that may reflect stochastic degradation rather than regulated gene silencing [82]. These complexities necessitate stringent validation criteria rather than accepting degradome signals at face value. High-confidence miRNA target validation requires convergent evidence from multiple independent lines [13,51].

Specifically, computational complementarity must be satisfied: the miRNA–target pair should typically permit no more than four mismatches under appropriate scoring thresholds as defined by the established criteria [79]. Precise degradome enrichment should be observed, with cleavage products showing peak abundance precisely at the 10th-11th nucleotide position of the miRNA–target duplex corresponding to the expected AGO cleavage site [13], where read abundance significantly exceeds background levels across the transcript. Category 0 or 1 peaks—where the cleavage signal represents the maximum or among the maxima on the transcript—offer the strongest support [13,14]. Where expression data are available, expression anti-correlation provides additional evidence, such that elevated miRNA expression coincides with reduced target mRNA accumulation in relevant tissues or conditions, consistent with miRNA-mediated regulation [83]. Finally, predicted interactions should exhibit biological plausibility, aligning with known expression patterns, tissue distribution, and functional contexts of both the miRNA and the target gene [2,3].

The PAREameters study quantitatively demonstrated these principles, showing that while computationally inferred criteria achieved 88.5% sensitivity compared to 81.4% for fixed Allen criteria, precision remained comparable at 91.3% versus 91.4% only when stringent degradome category filters—restricting analysis to categories 0, 1, and 2—were applied to exclude lower-confidence signals [51]. Thus, degradome sequencing provides powerful validation evidence when interpreted within a multi-layered framework that accounts for the biological complexity of cellular RNA metabolism [82]. However, isolated degradome peaks lacking supporting sequence complementarity and expression correlation must be interpreted with appropriate caution to avoid conflating coincidental cleavage with functional miRNA targeting [51,82]. Tests on three species—Arabidopsis, rice, and grape—showed that this integrated machine learning method significantly reduced false-positive rates, with accuracy improvements clearly surpassing single tools relying solely on sequence complementarity rules [81].

The development of degradome sequencing technology provided powerful experimental validation means for target gene prediction, enabling direct capture of mRNA cleavage products. The PAREameters tool (2020) is specifically designed to infer miRNA target prediction criteria from degradome data rather than applying fixed universal rules [51]. Traditional methods employ predetermined target identification criteria, such as allowed numbers of mismatches and gaps, typically based on limited data from Arabidopsis. The innovation of PAREameters resides in its capability to infer optimal criteria from users’ own small RNA and degradome datasets, using Bayesian inference and machine learning methods to learn miRNA–target pairing rules directly from experimental data.

Testing on multiple Arabidopsis datasets and non-model plants revealed that data-driven criteria identified more true interactions than fixed criteria, improving both sensitivity and accuracy of target prediction. This proves especially valuable for non-model plants because subtle differences may exist in miRNA–target recognition rules among different plant species, reflecting evolutionary divergence in regulatory mechanisms [51]. The smartPARE tool (2021) provides a user-friendly R package for efficiently identifying true mRNA cleavage sites and integrating significant cleavage peaks in degradome data with miRNA prediction results, streamlining the validation workflow [52]. The package is available through GitHub (https://github.com/KristianHoden/smartPARE, accessed on 22 November 2025) or CRAN (users should verify the current repository location and installation requirements).

Although deep learning has been widely applied in miRNA identification, as detailed in previous sections, its application in plant miRNA target gene prediction remains relatively limited. This may relate to several factors: the limited number of experimentally validated miRNA–target pairs available for training large neural networks, the inherently pairwise nature of target prediction, making it conceptually distinct from classification problems, and the need for clear biological interpretation of pairing positions and complementarity patterns, where the black-box nature of deep learning presents obstacles [84]. However, some studies have explored deep learning applications in related tasks. Zhang and colleagues’ use of CNN combined with IndRNN to predict miRNA–lncRNA interactions, and Kang and colleagues’ ensemble deep learning method, demonstrate approaches potentially extendable to miRNA–mRNA target prediction [55,58].

Future potential application directions for deep learning in target prediction include using Transformers to model long-distance complementary relationships between miRNA and target mRNA sequences, identifying critical pairing sites through attention mechanisms that highlight which positions contribute most to binding, integrating multi-modal data such as sequence complementarity combined with structural accessibility and expression correlation, and constructing miRNA–target interaction networks based on graph neural networks that capture regulatory network topology [68,85]. As training datasets expand through continued degradome sequencing and experimental validation efforts, deep learning methods may increasingly contribute to target prediction by learning subtle patterns distinguishing functional from spurious complementarity.

### 4.2. Exploration of Multi-Omics Data Integration

The functional regulation exerted by plant miRNAs constitutes a complex multi-level process, and single omics data types often prove insufficient to fully reveal their mechanisms of action. Multi-omics integration approaches attempt to depict miRNA biological functions from different perspectives by jointly analyzing genomic, transcriptomic, degradome, proteomic, metabolomic, and other data types [86].

Integrating miRNA expression profiles derived from small RNA sequencing with mRNA expression profiles from RNA sequencing represents the most fundamental strategy. By identifying miRNA-mediated negative regulatory relationships—where miRNA high expression correlates with target gene low expression—researchers can discover tissue-specific or developmental stage-specific miRNA–target pairs, inferring miRNA functions in particular biological processes. However, this integration faces substantial challenges. Data heterogeneity refers to significant differences in format, dimensions, and noise levels across different omics data types, requiring sophisticated normalization and integration methods. Spatiotemporal mismatch acknowledges that miRNA regulation exhibits temporal lags, where mRNA degradation and translational repression may be asynchronous in time and space, complicating correlation-based inference. Indirect effects arise because miRNAs may affect other genes indirectly by regulating transcription factors, creating regulatory cascades that obscure direct targets [87].

Degradome sequencing, as a technology specifically designed for identifying miRNA cleavage targets, directly provides experimental evidence of miRNA–target interactions by capturing mRNA fragments bearing 5′ monophosphate termini characteristic of AGO protein-mediated cleavage [13]. The technology enriches RNA fragments possessing 5′ phosphate termini, performs high-throughput sequencing of these enriched populations, maps sequencing reads to the transcriptome where read accumulation peaks correspond to cleavage sites, and combines results with miRNA sequence alignment to confirm miRNA–target pairs with strong evidence [14]. This direct experimental validation substantially improves confidence in predicted interactions compared to purely computational approaches.

Network-based analysis methods reveal systemic regulatory roles of miRNAs by recognizing that these molecules typically function within complex regulatory networks rather than in isolation. MicroRNAs interact with other genes, transcription factors, and signaling pathways through intricate regulatory networks. Network analysis enables identification of core gene modules regulated by miRNAs, discovery of synergistic regulation between miRNAs and transcription factors, and prediction of miRNA functional roles within specific biological pathways [88].

Although multi-omics integration research in the plant miRNA field remains relatively limited compared to animal systems, primarily focused on two-layer integration of miRNA omics and transcriptomics, several promising future opportunities exist. Multimodal deep learning approaches could develop models that analyze multiple data modalities simultaneously, including sequences, structures, expression profiles, and phenotypes within unified frameworks. Spatiotemporal dynamic modeling could integrate time series data from different developmental stages and spatial data from single-cell or spatial transcriptomics to reveal spatiotemporal dynamics of miRNA regulation across tissues and developmental transitions. Causal inference methods could distinguish direct regulatory relationships from indirect correlations in miRNA–target interactions, constructing causal networks that more faithfully represent biological causality rather than mere association. Pre-trained foundation models could be developed on large-scale multi-omics datasets, and then fine-tuned for specific prediction tasks, leveraging transfer learning to overcome data limitations in individual studies [89,90]. These multi-omics integration strategies become particularly powerful when applied to ceRNA network analysis, where multiple layers of regulatory information must be synthesized to understand complex RNA–RNA interactions, as discussed in the following section.

### 4.3. Competing Endogenous RNA Networks: Another Dimension of Regulation

Before examining miRNA interactions with long non-coding RNAs in the context of competing endogenous RNA networks, fundamental ambiguities surrounding the lncRNA definition and classification warrant acknowledgment. Long non-coding RNAs are operationally defined as transcripts exceeding 200 nucleotides that lack substantial protein-coding capacity [91]. Yet this definition proves problematic because the 200-nucleotide threshold represents an arbitrary boundary rather than a functional distinction reflecting biological reality [92], and the criterion of non-coding capacity proves difficult to assess definitively given that some annotated lncRNAs contain short open reading frames of uncertain translation status [93]. Furthermore, functional characterization of plant lncRNAs substantially lags behind their computational identification, with thousands of candidate lncRNAs annotated across plant transcriptomes but only a small fraction having experimentally validated regulatory functions [12]. This raises legitimate concern that many putative lncRNAs may represent transcriptional noise rather than functional regulatory molecules [94].

Given this fundamental uncertainty about which lncRNAs are biologically relevant, computational predictions of miRNA–lncRNA interactions face compounded challenges: not only must the interaction itself be validated, but the functional significance of the lncRNA must also be established. This necessitates that computational predictions be supported by robust experimental evidence demonstrating both the interaction and its biological consequences, as exemplified by well-characterized systems like the *IPS1*–*miR399* interaction in phosphate starvation responses [95,96]. These considerations underscore the importance of maintaining appropriate skepticism regarding miRNA–lncRNA interaction predictions and the necessity of rigorous experimental validation before drawing biological conclusions.

The competing endogenous RNA hypothesis posits a regulatory mechanism with broad implications [97]. The core concept holds that all RNA molecules containing miRNA binding sites—including mRNAs, long non-coding RNAs, circular RNAs, and pseudogene transcripts—can indirectly regulate each other’s expression levels through competitive binding of shared miRNAs. The basic mechanism operates as follows: when RNA molecule A and RNA molecule B both contain binding sites for miRNA X, increased expression of RNA A will sequester more miRNA X molecules, reducing available miRNA X, thereby diminishing repression on RNA B and increasing its expression. This creates an indirect positive regulatory relationship between A and B mediated through their competition for the shared miRNA [98].

Although the ceRNA hypothesis originated in animal systems, ceRNA networks also exist widely in plants and participate in multiple biological processes, including growth and development regulation, hormone signal transduction, and secondary metabolism regulation [99]. Constructing ceRNA networks requires several analytical steps: identifying miRNA binding sites using target prediction tools to map all possible miRNA–RNA interactions, calculating the number of shared miRNAs by counting how many miRNAs are predicted to target different RNA molecule pairs, evaluating competition intensity by considering factors, including the number of miRNA binding sites per transcript, binding affinity reflected in complementarity scores, and relative expression abundance of competing RNAs, and performing expression correlation validation to confirm that ceRNA pairs exhibit positively correlated expression patterns as predicted by the competition hypothesis [100].

Research published in 2021 specifically targeted mining of plant endogenous target mimics, a specialized form of ceRNA with substantial experimental validation [101]. In plants, long non-coding RNAs can function as endogenous target mimics, acting as molecular “decoys” that mimic miRNA binding sites of genuine target genes, thereby competitively binding miRNAs and relieving miRNA repression on authentic target genes.

The biological significance of this mechanism was first demonstrated in phosphate starvation responses, where the lncRNA IPS1 (Induced by Phosphate Starvation 1) functions as a target mimic for *miR399*, preventing *miR399* from cleaving its target *PHO2*—a ubiquitin-conjugating E2 enzyme—thereby maintaining phosphate homeostasis under phosphate-limiting conditions [95]. This pioneering discovery established the experimental paradigm for validating eTM functions through multiple complementary approaches. The validation framework begins with demonstrating sequence complementarity between the eTM and miRNA, except for a central bulge that prevents endonucleolytic cleavage while maintaining binding affinity, followed by characterizing co-expression patterns consistent with competitive interaction where eTM expression inversely correlates with miRNA activity on authentic targets. Additionally, the paradigm requires showing that eTM overexpression phenocopies miRNA loss-of-function, creating similar developmental or physiological effects as miRNA depletion, while genetic manipulation experiments must confirm that the observed eTM effect depends on the presence of the miRNA binding site rather than other eTM sequence features [96].

This study adopts a novel ensemble learning strategy of dual-path parallel ensemble pruning for predicting miRNA–lncRNA interactions, implementing an adaptive pruning mechanism that selects base learners based on model performance to improve ensemble efficiency. The model demonstrates applicability to different plant species without requiring retraining, facilitating cross-species prediction for organisms with limited training data.

The important contribution of this research lies in systematic experimental validation and functional inference bridging computational prediction with biological reality. From predicted miRNA–lncRNA interactions, potential endogenous target mimics were identified and subjected to rigorous experimental validation across multiple complementary approaches. Specifically, 17 potential eTMs were validated through comprehensive expression analysis across multiple tissues and stress conditions, revealing that these eTMs participated in 22 distinct regulatory relationships with specific miRNAs and their target genes. The validation strategy combined expression pattern analysis showing an inverse correlation with miRNA-targeted genes, sequence analysis confirming characteristic central mismatches that prevent AGO-mediated cleavage while maintaining binding capacity, and co-expression network analysis demonstrating regulatory connections consistent with competitive endogenous RNA function. Through Gene Ontology enrichment analysis, 14 eTMs were functionally annotated to participate in 63 biological processes spanning stress responses, hormone signaling pathways, and developmental processes [101]. These findings revealed the important role of plant long non-coding RNAs as endogenous target mimics in gene regulatory networks, providing computational tools and experimental data for understanding the three-layer regulatory architecture encompassing miRNAs, lncRNAs, and mRNAs. Predicted eTMs serve as potential targets for crop improvement applications, as manipulation of eTM expression levels could enable fine-tuning of miRNA activity without directly modifying miRNA genes themselves, offering a new avenue for precision breeding strategies.

The PceRBase database (2017) represents the first comprehensive plant ceRNA database [102]. This resource (http://bis.zju.edu.cn/pcernadb/index.jsp, accessed on 22 November 2025; users should verify current accessibility) includes ceRNA network-related data from 26 plant species (later expanded to 28), encompassing miRNAs, target genes, long non-coding RNAs, and inferred ceRNA relationships. The database received its last documented update in October 2018 with the addition of *B. napus* and *H. annuus*; users should be aware that the maintenance status since 2018 is unclear, and current functionality should be verified before relying on this resource for research projects. For Arabidopsis, as an example, 311 candidate ceRNAs were identified, potentially affecting 2646 target–miRNA–target regulatory interactions. The database stores predicted pairing structures between miRNAs and their target mRNA transcripts, expression levels of ceRNA pairs derived from publicly available RNA-seq datasets, and associated Gene Ontology annotations linking ceRNAs to biological functions. The web interface provides convenient browsing and search functions, along with tools for ceRNA network visualization and enrichment analysis. Users can also employ PceRBase to predict novel competing mimic–target and target–target interactions in their own datasets by applying the database’s computational pipelines [102].

Current challenges in ceRNA research include validation difficulties, as ceRNA relationships are typically indirect and experimental validation requires simultaneous manipulation of multiple molecular species with high technical demands. Dynamic complexity arises because ceRNA interaction strength depends on relative expression abundances and may vary dramatically across different tissues, developmental stages, and environmental conditions. False-positive risk remains substantial, as predicting ceRNAs based solely on shared miRNA binding sites and expression correlation easily produces spurious relationships that lack biological functionality. Difficulty in quantitative modeling persists, as accurately modeling quantitative competitive relationships between miRNAs and ceRNAs in terms of binding kinetics and stoichiometry remains challenging [103,104].

Future development directions for ceRNA research include integrating multi-layer data by combining protein–RNA interaction data from technologies like RIP-seq and CLIP-seq to more accurately identify functional miRNA binding sites rather than relying solely on sequence-based predictions. Constructing dynamic network models that capture time-dependent and condition-dependent ceRNA networks rather than static snapshots will better represent biological reality. Deep learning applications using graph neural networks and related methods to model complex ceRNA network topology and predict dynamic network evolution offer promising avenues. Systems biology integration that embeds ceRNA networks within larger regulatory networks encompassing transcription factors, chromatin modifications, and signaling pathways will provide a more complete understanding. Finally, developing high-throughput ceRNA functional validation methods, including massively parallel reporter assays and CRISPR-based perturbation screens, will accelerate experimental confirmation of computationally predicted relationships [105,106].

## 5. Challenges and Prospects

### 5.1. Data Quality and Annotation Standards

Although AI research on plant miRNAs has achieved significant progress, fundamental challenges in data quality and consistency continue to limit the reliability of computational predictions. The heterogeneity of annotation standards across studies represents a critical obstacle that pervades the field. As established by comprehensive evaluations, approximately 75% of plant miRNA families in miRBase version 20 lacked convincing supporting evidence, with only 9.4% of land plant miRNA families designated as high-confidence in miRBase version 21 [7]. This alarming discrepancy arose primarily from inconsistent annotation criteria applied by different research groups and the minimal intervention approach historically adopted by miRBase curators [20].

To address these quality concerns, the plant miRNA community established updated annotation criteria in 2018, emphasizing specific requirements that modern tools should implement [7]. These standards mandate precise excision from stem-loop precursors with characteristic hairpin topology, accumulation of both mature miRNA and miRNA* sequences with the diagnostic 2-nucleotide 3′ overhang, and validation through independent small RNA sequencing datasets. Modern prediction tools now implement these standards through explicit confidence classification systems. For instance, PmiREN employs a three-tier confidence framework: high-confidence miRNAs must be detected using stringent criteria, including precise precursor processing, abundant expression with reads per million ≥10, and structural validity; intermediate-confidence entries show lower expression but meet structural requirements; and low-confidence annotations satisfy only structural criteria without robust expression support [20]. Similarly, miRScore implements quantitative thresholds requiring at least 75% precision in read alignment, enforcing the canonical 2-nucleotide 3′ overhang in miRNA/miRNA* duplexes, and validating expression levels above background noise [49].

Different tools assign substantially different weights to these criteria based on their design philosophy. Sequence-based predictors typically prioritize stem-loop secondary structure stability measured by minimum free energy with thresholds of approximately −20 kcal/mol and base-pairing patterns requiring at least 16 paired nucleotides in stem regions, while expression-based tools weight read abundance and precision more heavily. Machine learning tools like PlantMiRNAPred assign feature importance through training processes, with structure-related features, including stem length and loop size, along with thermodynamic features, such as normalized minimum free energy, consistently ranked among the top discriminative characteristics [22,24]. Understanding these criteria and their relative weights proves essential for interpreting prediction results and assessing confidence levels appropriately.

Data quality varies significantly across sources: early miRNAs identified through cloning sequencing may harbor annotation errors, high-throughput sequencing data, while abundant, contain substantial noise, and different studies employ inconsistent experimental conditions, sequencing platforms, and analysis pipelines, all contributing to data heterogeneity that AI models must somehow accommodate [20]. This heterogeneity fundamentally limits model performance, as training on noisy or inconsistent labels inevitably produces models that learn artifacts alongside genuine biological patterns.

### 5.2. Non-Model Plant Data Scarcity: A Critical Bottleneck

The disparity in miRNA data availability between model and non-model plant species represents one of the most significant obstacles to achieving a comprehensive understanding of plant miRNA biology. While model organisms like *Arabidopsis thaliana* and *Oryza sativa* benefit from decades of accumulated data, including hundreds of validated miRNA loci supported by multiple independent studies, recent years have witnessed nearly 200 plant genomes newly sequenced, yet corresponding miRNA annotations lag far behind [107]. This data imbalance manifests across multiple dimensions with profound implications for computational method development and application.

Genomic resource disparity creates a vicious cycle that perpetuates inequality in knowledge generation. Model plants possess complete genome assemblies, extensive transcriptome datasets, and thousands of small RNA sequencing libraries, enabling robust miRNA discovery and characterization. Conversely, many economically important crops and wild relatives lack even basic genomic resources [47]. While Arabidopsis boasts over 400 validated miRNA loci with comprehensive functional annotations, many orphan crops possess fewer than 50 computationally predicted miRNAs without experimental validation. This creates a self-reinforcing pattern: without genomic references, homology-based miRNA prediction fails; without predicted miRNAs, researchers lack targets for experimental validation; and without validation data, securing funding for sequencing projects becomes difficult, perpetuating the resource gap.

Species-specific biology challenges emerge because the fundamental assumption underlying transfer learning and cross-species prediction—that miRNA families are evolutionarily conserved—breaks down for species-specific or recently evolved miRNAs. Non-conserved miRNAs, often expressed at low levels and potentially regulating lineage-specific adaptations, remain systematically under-detected in non-model species due to insufficient sequencing depth and lack of species-optimized prediction parameters [7]. For instance, stress-responsive miRNAs in halophytes adapted to extreme salinity or temperature-adaptive miRNAs in alpine plants may have no homologs in model organisms, rendering homology-based discovery methods completely ineffective. These species-specific regulatory innovations potentially hold keys to understanding environmental adaptation mechanisms, yet they remain largely hidden from current computational approaches.

Experimental validation barriers compound these challenges even when computational predictions are generated for non-model species. Transformation protocols may be unavailable or inefficient for many species, generation times may span years rather than months, precluding rapid phenotypic analysis, and tissue culture recalcitrance prevents functional studies in numerous economically important species. The recommendation to require at least 10 matched reads for confident miRNA validation in animals must be relaxed to as few as three reads in plants due to the relative scarcity of plant small RNA sequencing experiments [47], but this compromise introduces uncertainty about distinguishing true miRNAs from spurious low-abundance transcripts.

Several computational solutions have been proposed to mitigate data scarcity. Transfer learning approaches that pre-train models on data-rich species, followed by fine-tuning on limited target species data, have shown promise, achieving accuracy improvements of 15–30% in various plant classification tasks when source and target domains share biological similarities [70]. However, effectiveness critically depends on phylogenetic distance: transferring from Arabidopsis to rice works moderately well, but transferring to algae or bryophytes often fails due to fundamental differences in miRNA biogenesis machinery and target recognition mechanisms. Tools designed for resource-limited scenarios have also emerged. The microRPM tool pioneered genome-independent prediction using only small RNA-seq data, successfully identifying and validating 18 out of 21 predicted orchid miRNAs without requiring a reference genome [26]. Similarly, the miRPreM method adopts organism-specific reference strategies rather than genome-centric approaches, more than doubling the discovery rate in resource-limited species [27]. However, these methods sacrifice some accuracy for broader applicability, with precision typically 5–10% lower than genome-informed approaches when both are feasible.

Emerging machine learning paradigms, including few-shot learning and data augmentation through synthetic sequence generation, offer potential solutions but remain largely unexplored in plant miRNA research. Few-shot learning could enable model training from as few as 5 to 10 examples per species by learning general principles of miRNA structure from many species that transfer to new organisms with minimal data. Data augmentation could artificially expand training sets by generating synthetic miRNA precursor sequences that respect biological constraints of secondary structure and thermodynamic stability. Proof-of-concept studies demonstrating feasibility in plant miRNA contexts are needed before practical deployment.

Addressing data scarcity ultimately requires coordinated community efforts beyond individual laboratories. Initiatives like PmiREN2.0, which systematically re-annotated miRNAs across 91 species using standardized pipelines [107], exemplify the value of centralized, quality-controlled resources. However, sustainability remains challenging: Who maintains these databases long-term once initial funding expires? How are updates funded as new sequencing data accumulates? What incentives encourage researchers to contribute data rather than hoard it for exclusive publication rights? These socio-scientific questions may ultimately prove as important as technical algorithmic advances in determining whether the field can overcome data scarcity limitations.

### 5.3. Model Interpretability Challenges

Deep learning models may outperform traditional methods in prediction performance, but their black-box nature creates a fundamental tension with biology research’s need for mechanistic understanding [71]. Biologists require not merely accurate predictions but insight into why particular sequences are classified as miRNAs, which features prove most critical for miRNA biogenesis and function, and what testable biological hypotheses emerge from model behavior. Computational predictions should guide experimental design rather than replacing biological intuition [72].

The nucleotide importance visualization provided by mirDNN represents one approach to enhancing interpretability [54]. This tool identifies sequence regions most critical for predictions by visualizing model attention weights, enabling researchers to examine which nucleotide positions most strongly influence classification decisions. However, attention does not equal causation: regions receiving high attention are not necessarily biologically critical regions. The fundamental challenge persists that even when convolutional filter-learned patterns can be visualized, interpreting these visualizations still requires domain expertise, and while shallow convolutional kernels may capture interpretable motifs, deep abstract features often resist straightforward biological interpretation [75].

Beyond interpretability limitations, experimental validation presents substantial practical challenges. Even when models provide some mechanistic explanation, translating insights into verifiable biological hypotheses and conducting rigorous experimental validation remains time-consuming and resource-intensive. Designing mutant constructs to validate the functional role of specific nucleotide positions requires constructing expression vectors, transforming plant cells, culturing transformants, and conducting phenotypic observations, with experimental cycles ranging from months to years, depending on the species. Comprehensive functional research demands combinations of multiple experimental techniques, including Northern blotting, quantitative RT-PCR, degradome sequencing, and phenotypic analysis, creating significant bottlenecks for high-throughput validation of computational predictions [108].

The path forward requires developing interpretable AI approaches specifically designed for biological sequence analysis. Attention mechanisms can be enhanced to provide more biologically meaningful visualizations, SHAP values can quantify individual feature contributions to predictions in model-agnostic ways, and counterfactual analysis can identify critical sequence elements by systematically perturbing inputs and observing prediction changes. However, these technical solutions must be coupled with experimental validation frameworks that enable efficient testing of computational hypotheses at scale.

### 5.4. Complex Regulatory Network Modeling

Plant miRNA regulatory networks exhibit multi-level, multi-scale, and dynamically changing characteristics that pose fundamental modeling challenges resisting simple computational solutions. At the multi-level dimension, these networks span transcriptional regulation of miRNA genes by transcription factors and chromatin modifications, post-transcriptional miRNA processing and maturation involving the *DCL1* complex and accessory proteins, target-level interactions, including miRNA-mRNA and miRNA-lncRNA regulation, translational repression mechanisms distinct from mRNA cleavage, and ceRNA-level competitive interactions creating indirect regulatory relationships [109]. The multi-scale nature encompasses molecular-level phenomena, including individual miRNA–target pairing dynamics, pathway-level integration within signaling cascades where miRNAs modulate responses to hormones and stress, network-scale topology and dynamics determining global regulatory architecture, and system-scale effects on organismal growth, development, and metabolic efficiency [110].

Current modeling approaches face several critical limitations that constrain their ability to capture this biological complexity. Data heterogeneity integration remains challenging, as different omics data types possess vastly different formats, dimensions, and noise characteristics, lacking effective multimodal integration frameworks that can optimally combine information while accounting for source-specific biases and uncertainties. Static model dominance persists, with most approaches constructing static networks that ignore the dynamic temporal evolution of regulatory relationships across developmental stages and environmental conditions. Scalability bottlenecks emerge when genome-wide networks contain thousands of nodes and tens of thousands to hundreds of thousands of edges, resulting in computational complexity that renders comprehensive network inference and simulation intractable. Validation difficulties arise because network model validation requires large amounts of experimental data characterizing many regulatory relationships simultaneously, while experimental validation of global networks proves practically infeasible, limiting confidence in predicted network structures [111,112].

### 5.5. Future Development Directions

Addressing the challenges outlined above requires strategic advances across multiple fronts, from technical innovations in algorithm design to systematic improvements in data quality and experimental validation frameworks (Figure 4). The following directions represent the most promising pathways forward for advancing AI applications in plant miRNA research.

Multimodal deep learning will become an increasingly important direction as the field recognizes that plant miRNA function is determined by multiple factors operating synergistically: sequence composition dictating target specificity, secondary and tertiary structure enabling biogenesis and RISC loading, chromatin accessibility controlling transcription, expression levels determining regulatory impact, tissue and developmental stage distribution specifying functional context, and evolutionary conservation indicating functional importance. Single-modal data often proves insufficient to fully characterize miRNAs and their functions [113]. Technical frameworks for multimodal integration include hybrid fusion strategies combining early fusion that concatenates features before modeling with late fusion that ensembles predictions from modality-specific models, attention mechanisms that dynamically adjust weights of different modalities based on their informativeness for specific predictions, and architectures, including multi-stream networks processing each modality separately before integration and cross-modal attention learning relationships between different data types [114].

Potential applications span multiple domains. Integrating sequence and structure information could enable models to learn synergistic sequence–structure patterns where particular sequence motifs confer function only in specific structural contexts. Integrating expression and phenotype data could associate miRNA expression profiles reflecting dynamic behavior with phenotypic data linking miRNAs to functional traits, enabling identification of functionally important miRNAs beyond mere prediction of existence. Integrating evolutionary and functional information could leverage conservation patterns hinting at functional importance, homology with model organisms, enabling transfer of functional annotations, while identifying species-specific miRNAs potentially involved in adaptive evolution [115,116].

Deepening applications of explainable artificial intelligence will address black-box problems that currently limit biological insight extraction from high-performing models. In plant miRNA research, interpretability represents not merely a desirable feature but an essential requirement: biologists need to understand mechanisms behind predictions to gain scientific insights rather than treating models as oracles, biological plausibility of model decisions affects their credibility and application value in guiding experimental design, and interpretable models facilitate integration with existing knowledge while enabling discovery of novel hypotheses that extend current understanding [117].

Potential applications of explainable AI technology in miRNA research include SHAP values based on game-theoretic Shapley values to quantify each feature’s contribution to predictions, identifying sequence positions or structural features most important for miRNA recognition. This model-agnostic approach applies to any prediction model, though with substantial computational cost [118]. Deepening applications of attention mechanisms could reveal that different attention heads in multi-head attention architectures focus on different biological features, such as conserved regions, stem structures, or loop configurations, while hierarchical attention could reveal organizational principles from low-level sequence motifs to high-level functional modules [119]. Counterfactual explanations answer “what if” questions by examining how predictions change when specific inputs are modified, directly identifying critical nucleotides and corresponding to experimental manipulations like site-directed mutagenesis [120].

Tools such as PmlIPM and PmiProPred have already demonstrated the potential of attention mechanisms in enhancing interpretability [60,61]. Multi-head attention visualization enables examination of sequence regions receiving model focus, providing clues for understanding miRNA recognition mechanisms. Future directions include developing plant miRNA-specific explainable AI methods tailored to the biological constraints and patterns unique to plant systems, associating model explanations with known biological knowledge, including conserved motifs and structural rules to ground interpretations in established understanding, and using model-discovered patterns to guide new experimental hypotheses that test predictions in empirical studies [121,122].

Although graph neural network applications in plant miRNA research remain limited, they demonstrate tremendous potential for future development. RNA secondary and tertiary structures, miRNA regulatory networks, and ceRNA networks can all be naturally represented as graphs where nodes correspond to molecules and edges represent interactions, making GNNs an ideal architectural choice for processing these inherently relational data types [123]. Future application directions include spatiotemporal dynamic network models, developing methods that can explicitly model network temporal evolution, such as temporal graph neural networks, while integrating spatial transcriptome data to construct spatially aware regulatory networks revealing tissue-specific regulatory architectures [124]. Multi-layer heterogeneous network approaches could establish networks, including multiple node types, such as miRNAs, mRNAs, lncRNAs, proteins, and metabolites, employing heterogeneous graph neural networks specifically designed for networks with diverse node and edge types [125]. Causal network inference could distinguish correlation from causality through integration of perturbation experimental data, constructing causal regulatory networks that more faithfully represent biological causality rather than mere statistical association [126].

Although current AI applications specifically for plant miRNA de novo design remain extremely limited, this direction holds long-term potential as computational and experimental capabilities mature. De novo design of functional miRNAs represents the ultimate application of AI in this field, moving beyond passive prediction to active creation of regulatory elements with specified properties. Several technical pathways are emerging as potentially viable approaches, though substantial development remains necessary before practical deployment.

First, generative adversarial networks could potentially be adapted from their demonstrated success in protein design to generate novel miRNA precursor sequences satisfying both structural constraints, including stable stem-loop formation and appropriate thermodynamic properties, and functional requirements, encompassing target specificity and processing compatibility with the DCL1 machinery. The discriminator network would be trained on validated plant miRNAs to distinguish realistic miRNA sequences from random hairpins, while the generator learns to create sequences that fool the discriminator by exhibiting authentic miRNA characteristics. Second, variational autoencoders offer a probabilistic framework for sampling the latent space of miRNA sequence–structure relationships, enabling exploration of sequence variants with desired properties while maintaining biological plausibility by constraining generation to regions of latent space corresponding to viable miRNA structures. Third, and perhaps most promisingly for near-term applications, fine-tuning large RNA foundation models, such as plant-specific variants of RNA-FM or RNA-MSM, could leverage transfer learning from millions of plant RNA sequences to guide design of miRNAs with specified target profiles or expression patterns [127,128]. Early proof-of-concept work in animal systems has demonstrated that attention-based language models can learn syntactic rules of miRNA–target interactions directly from sequence data without explicit programming of complementarity rules [129].

However, substantial obstacles must be overcome before AI-driven de novo miRNA design becomes practically viable. Plant miRNA design faces multi-constraint optimization challenges requiring simultaneous satisfaction of thermodynamic stability with minimum free energy of approximately −25 kcal/mol, target specificity minimizing off-target effects through careful sequence selection, processing compatibility ensuring DCL1 recognition and precise cleavage, and expression efficiency depending on appropriate promoter selection and chromatin context. Current models typically handle one or two constraints but not all simultaneously [127]. The validation bottleneck proves even more severe: plant transformation requires 6–12 months from construct design to phenotypic analysis, no high-throughput assay exists for rapidly testing miRNA function in planta, and epistatic effects in complex plant genomes create context-dependencies that resist simple prediction. This prevents the rapid design–test iteration cycles that enabled success in protein design, where computational predictions can be validated within days through synthesis and in vitro assays. Training data scarcity further constrains current approaches, as fewer than 500 high-confidence plant miRNAs per species provide insufficient training data for generative models that typically require thousands to tens of thousands of examples to learn meaningful patterns. Transfer learning from animal miRNAs fails due to fundamental mechanistic differences in biogenesis and target recognition.

Realistic near-term goals through 2030 should prioritize more tractable objectives rather than full de novo generation. Variant optimization proves feasible: starting with validated miRNA scaffolds, reinforcement learning could optimize target specificity, ensemble methods could predict off-targets, with expected success rates exceeding 80%. Target site selection represents another achievable goal: given a target gene, algorithms could predict optimal binding sites, considering secondary structure accessibility and minimizing SNP-sensitive regions, with expected success rates above 70%. In contrast, full de novo generation, creating entirely novel miRNAs from scratch with arbitrary target gene specificity and guaranteed functionality, remains currently infeasible, with realistic timelines extending to 2030–2035 or beyond [127,130].

In the short term, spanning the next 1–3 years, transfer learning using general biological sequence models can be explored, specialized plant RNA language models can be developed, capturing plant-specific regulatory grammar, and AI can assist literature mining and knowledge integration to accelerate hypothesis generation. In the medium term spanning 3–5 years, multimodal AI models could be constructed integrating multi-source information, including sequences, structures, expression patterns, and literature-derived functional annotations, while interactive miRNA research assistants could be developed to aid experimental design and data analysis. In the long term, spanning 5–10 years, de novo design of miRNAs with specific functions may be achieved if validation throughput increases dramatically through technological breakthroughs, general foundation models for plant genomics could be constructed, enabling zero-shot prediction across species, and AI-driven autonomous scientific discovery systems might emerge, though this remains highly speculative [130].

**Figure 4 ijms-26-11854-f004:**
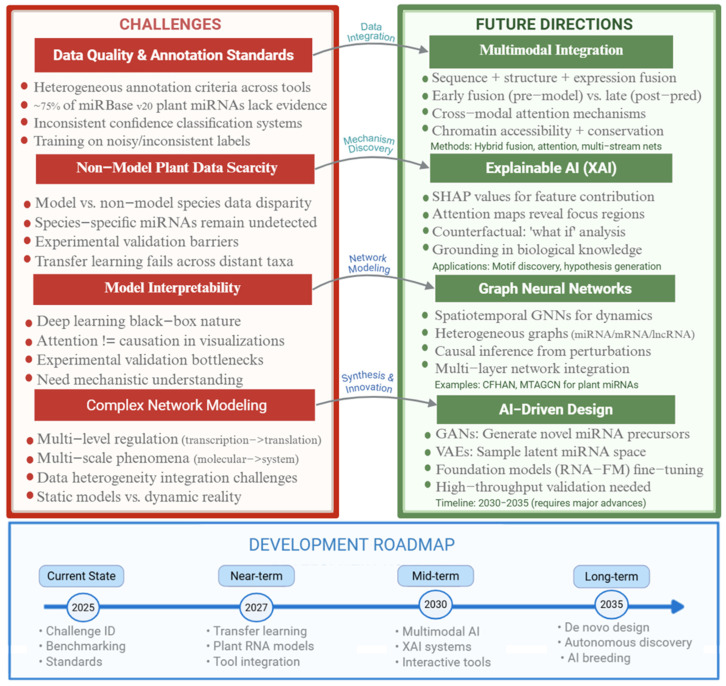
Strategic Framework for Addressing Challenges in AI-Driven Plant miRNA Research. This comprehensive framework maps current limitations to prospective solutions in plant miRNA computational research, structured as a challenge-solution paradigm with implementation timelines. The left panel (red background) delineates four critical challenges corresponding to Section 5.1, Section 5.2, Section 5.3 and Section 5.4: data quality and annotation standards (Section 5.1), including heterogeneous criteria and insufficient evidence for ~75% of miRBase v20 entries; non-model plant data scarcity (Section 5.2) characterized by species imbalance and transfer learning failures across distant taxa; model interpretability issues (Section 5.3) stemming from black-box neural architectures where attention does not equal causation; and complex network modeling difficulties (Section 5.4) involving multi-level regulation and static versus dynamic modeling trade-offs. The right panel (green background) presents four future research directions from Section 5.5: multimodal integration combining sequence, structure, and expression data through hybrid fusion strategies; explainable AI employing SHAP values and attention mechanisms for transparent predictions; graph neural networks (exemplified by CFHAN and MTAGCN) capturing spatiotemporal dynamics; and AI-driven de novo design leveraging GANs, VAEs, and foundation models. Curved arrows with conceptual labels (Data Integration, Mechanism Discovery, Network Modeling, Synthesis & Innovation) connect each challenge to its targeted solution, indicating strategic pathways for methodological advancement. The bottom panel (blue background) provides a staged roadmap spanning 2025 (current challenges and benchmarking), 2027 (near-term solutions, including transfer learning), 2030 (mid-term multimodal AI and XAI systems), to 2035 (long-term autonomous discovery and breeding applications), offering temporal context for the evolution of computational methodologies over the next decade.

## 6. Conclusions

This review has systematically examined the current application status, persistent challenges, and future development directions of artificial intelligence technology in plant microRNA research. Through comprehensive analysis of methodological advances and critical evaluation of achievements and limitations, several main conclusions emerge that define the current state of the field and illuminate paths forward.

Traditional machine learning methods have matured substantially in plant miRNA identification, classification, and target gene prediction, producing a suite of high-performance tools, including PlantMiRNAPred, miPlantPreMat, plantMirP, and others. These methods perform excellently on small sample datasets, where deep learning approaches often struggle due to data requirements, with feature importance analysis providing interpretability that supports hypothesis generation and biological understanding. However, the field has increasingly recognized that prediction accuracy alone proves insufficient without rigorous attention to annotation quality. The establishment of standardized criteria for plant miRNA annotation, including requirements for precise processing, characteristic structural features, and independent validation, has set new benchmarks that AI tools must meet to ensure reliable predictions rather than perpetuating false annotations accumulated in earlier database curation efforts.

The rapid development of deep learning technology has brought transformative breakthroughs to this field. Tools such as mirDNN significantly reduced false-positive rates in genome-wide scanning through sophisticated residual network architectures, while models, including CIRNN and PmliPEMG, achieved excellent performance in miRNA–ncRNA interaction predictions through hybrid architectures combining complementary network types. Deep learning’s automatic feature learning capability reduces dependence on manual feature engineering that dominated traditional approaches, providing fundamentally new approaches for discovering and modeling complex patterns in biological sequences.

At the application level, target gene prediction methods have evolved from sequence complementarity rules through machine learning enhancement to integration with degradome sequencing validation. Critical assessment of degradome data has revealed important limitations that must be considered in interpretation: while degradome sequencing provides valuable validation evidence for miRNA-mediated cleavage, the complexity of degradome datasets—including contributions from non-miRNA sources, technical artifacts, and background cleavage events—necessitates stringent filtering criteria and integration with computational predictions. The most reliable target validations combine sequence complementarity analysis, degradome evidence demonstrating precise cleavage at expected positions, and expression correlation across multiple datasets and conditions. The PCA-SVM ensemble method and degradome-focused tools like PAREameters represent important advances in achieving higher-confidence target predictions through multi-evidence integration.

Competing endogenous RNA network research has revealed additional layers of miRNA regulatory complexity, with systematic experimental validation of endogenous target mimic mechanisms demonstrating the functional importance of miRNA–lncRNA interactions in diverse biological processes spanning stress responses, hormone signaling, and developmental transitions. The PceRBase database and experimentally validated eTM examples provide concrete evidence supporting computational predictions and establish foundations for systems biology approaches to understanding integrated regulatory networks. Although frontier technologies such as graph neural networks have seen limited application to date, tools, including CFHAN and MTAGCN, have demonstrated their unique advantages in processing complex network-structured data, suggesting substantial future potential.

Current research faces multiple persistent challenges that constrain progress. Data quality and quantity limitations represent the most fundamental impediment, with species data imbalance creating a divide between well-characterized model organisms and agriculturally important non-model species, inconsistent annotation standards across tools and databases undermining confidence in training data, and experimental validation bottlenecks restricting high-throughput confirmation of computational predictions. Different tools employ varied annotation criteria with substantially different weight assignments: structure-based methods prioritize thermodynamic stability and base-pairing patterns, and expression-based methods emphasize read abundance and processing precision, while machine learning approaches learn feature importance empirically from training data with performance heavily dependent on training set quality. Insufficient interpretability of deep learning models hinders their application in hypothesis-driven biological research, where understanding mechanisms proves as important as achieving predictive accuracy. Complex regulatory network modeling faces challenges encompassing data heterogeneity, integration across omics types, dynamic modeling capturing temporal evolution, and scalability limitations for genome-wide analyses.

Looking toward the future, multimodal deep learning approaches integrating diverse data types, explainable artificial intelligence methods illuminating model decision processes, graph neural network applications for network topology analysis, and ultimately AI-based miRNA design represent promising directions for advancing the field. Emerging technical pathways for de novo miRNA design—including generative adversarial networks, variational autoencoders, and fine-tuned RNA foundation models—offer long-term promise for moving the field from passive discovery to active engineering of regulatory elements, though substantial obstacles in multi-constraint optimization, validation throughput, and training data availability must be overcome before practical deployment becomes feasible. These technologies will not only provide more efficient prediction tools but, more importantly, open new paradigms for discovering novel patterns from data and advancing mechanistic understanding of biological systems.

With the continued evolution of AI technology and the accumulation of plant biological knowledge, we have compelling reasons to believe that AI will play an increasingly important role in plant miRNA research and the broader field of plant science, contributing to solving pressing challenges in human food security and environmental sustainability. However, technological progress cannot be divorced from the essence of biological problems. Artificial intelligence should serve as a powerful tool advancing scientific discovery rather than becoming an end unto itself. We need to maintain clear understanding and appropriate skepticism: honestly evaluate performance metrics recognizing that high accuracy on test sets does not guarantee biological validity and acknowledging genuine limitations of both data sources and algorithmic assumptions; emphasize experimental validation recognizing that computational predictions must undergo rigorous experimental verification with awareness of technical limitations in validation approaches; emphasize interpretability recognizing that models must be not only accurate but understandable to generate biological insights and testable hypotheses; promote open science principles through sharing of code, data, models, and transparent documentation of annotation criteria and confidence levels; and focus on practical applications ensuring that AI advances translate to tangible benefits in crop improvement and agricultural sustainability rather than remaining purely academic exercises.

## Figures and Tables

**Figure 1 ijms-26-11854-f001:**
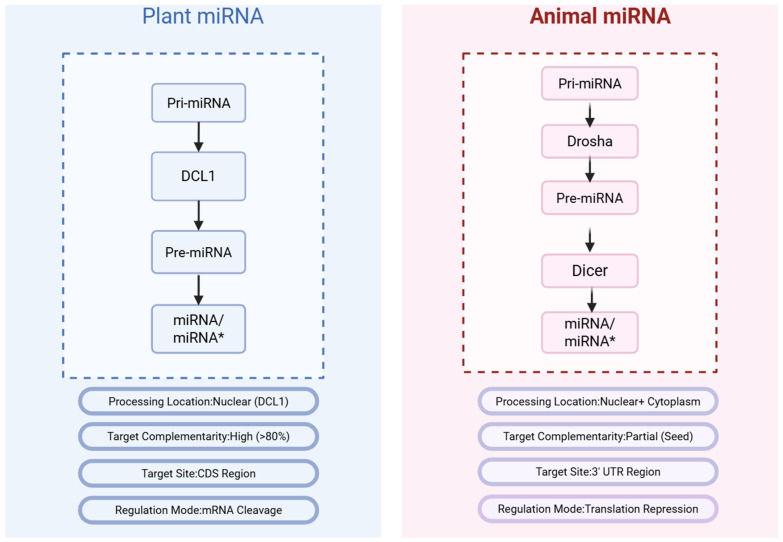
Fundamental Differences Between Plant and Animal miRNA Systems. This comparative diagram illustrates the distinct characteristics of plant and animal microRNA (miRNA) biogenesis pathways and regulatory mechanisms. The left panel (blue) depicts the plant miRNA system, showing nuclear-localized processing by DCL1 enzyme that converts pri-miRNA directly to pre-miRNA and subsequently to the mature miRNA/miRNA* duplex within the nucleus. The right panel (pink) presents the animal miRNA system, featuring a two-step processing mechanism involving nuclear Drosha-mediated cleavage of pri-miRNA to pre-miRNA, followed by cytoplasmic export and Dicer processing to generate the mature miRNA duplex. The lower section highlights four key distinguishing features between the two systems: processing location (nuclear for plants versus nuclear and cytoplasmic for animals), target complementarity (high complementarity exceeding 80% in plants versus partial seed-region matching in animals), target site preferences (coding sequence regions in plants versus 3′ untranslated regions in animals), and regulation mode (mRNA cleavage in plants versus translational repression in animals). These fundamental mechanistic differences underscore the evolutionary divergence of miRNA-mediated gene regulation across kingdoms and necessitate distinct computational approaches for its study.

**Figure 2 ijms-26-11854-f002:**
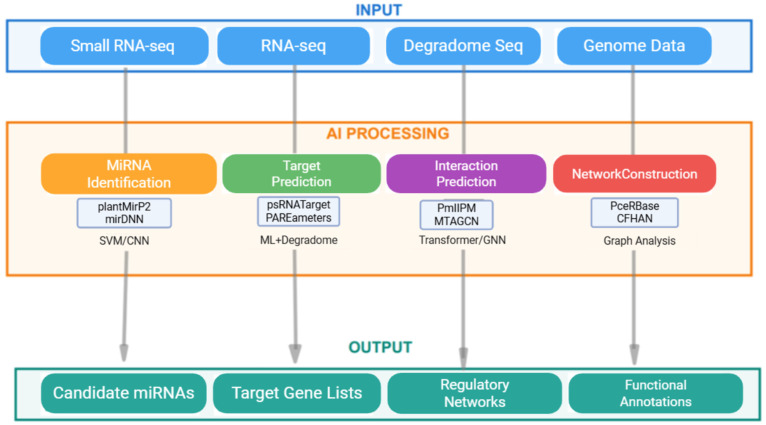
AI-Powered Pipeline for Plant miRNA Research. This workflow diagram presents a comprehensive artificial intelligence-driven pipeline for plant miRNA discovery and functional characterization, organized into three hierarchical layers. The input layer (top, blue background) encompasses diverse data sources, including small RNA sequencing, total RNA sequencing, degradome sequencing, and genome sequence data, which provide complementary information for miRNA analysis. The central AI processing layer (middle, orange background) consists of four parallel modules: miRNA identification utilizing plantMirP2 and mirDNN tools based on SVM and CNN architectures, target prediction employing psRNATarget and PAREameters with machine learning algorithms integrated with degradome evidence, interaction prediction leveraging PmlIPM and MTAGCN through transformer and graph neural network approaches, and network construction performed via PceRBase and CFHAN using graph analysis techniques. The output layer (bottom, teal background) delivers four categories of actionable results: candidate miRNA sequences for experimental validation, curated target gene lists for downstream functional studies, reconstructed regulatory networks revealing miRNA–target relationships, and functional annotations providing biological context. Connecting arrows indicate the directional flow of information processing, demonstrating how raw sequencing data are progressively transformed through AI-powered computational modules into biologically interpretable knowledge that bridges the gap between high-throughput sequencing and mechanistic understanding of plant miRNA regulatory systems.

**Table 1 ijms-26-11854-t001:** Criteria for miRNA Prediction Confidence Levels.

Criterion	Stringent (High-Confidence)	Moderate	Relaxed (Discovery)
MFE threshold	≤−25 kcal/mol	≤−20 kcal/mol	≤−18 kcal/mol
Stem paired bases	≥18 bp	≥16 bp	≥14 bp
Expression (RPM)	≥10	≥5	≥3
Read alignment precision	≥90%	≥80%	≥75%
miRNA* detection	Required	Preferred	Optional
2-nt 3′ overhang	Strict enforcement	Allowed deviation	Not enforced
Independent datasets	≥2	1	Computational only
Example tools	miRador [48], miRScore [49]	plantMirP2 [23]	miRDeep-P2 [21]

**Table 2 ijms-26-11854-t002:** Comparison of Deep Learning Architectures for miRNA Prediction.

Architecture	Tool	Key Parameters	Input	Training *	GPU Memory †	RAM	Accuracy
CNN	mirDNN [55]	3 conv blocks (64 → 128 → 256)	Seq + Struct	2–6 h	8–12 GB	16 GB	87–99%
CNN-RNN	CIRNN [56]	CNN 32 filters; IndRNN 64 units	Sequence	4–8 h	8–12 GB	8 GB	85–95%
LSTM	DIGITAL [58]	2 layers × 128 units	Sequence	3–5 h	4–8 GB	16 GB	94–98%
Transformer	PmlIPM [61]	8 heads; 128-dim; 6 layers	Seq pairs	12–24 h	16 GB ‡	32 GB	90–97%
GNN	CFHAN [64]	3 GCN + 3 attention	Graph	8–16 h	8–16 GB §	32–64 GB	97–99%

* RTX 3000-series GPU, batch size 32–128, on ~5–10 K training samples. † GPU VRAM for training. Inference needs ~30% of this. ‡ Minimum 16 GB; 8 GB possible with gradient accumulation. § Scales with graph: 4–6 GB (small), 12–16 GB (large).

## Data Availability

No new data were created or analyzed in this study. Data sharing is not applicable to this article.

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
