# Peer review of "Application of Artificial Intelligence Technology in Plant MicroRNA Research: Progress, Challenges, and Prospects"

_ijms, 2025, doi:10.3390/ijms262411854_

Round 1
Reviewer 1 Report
Comments and Suggestions for Authors
Plant microRNAs play critical roles in plant growth, development, and stress responses. The manuscript presents a comprehensive analysis of AI applications in plant microRNA research, encompassing progress, challenges, and future directions. It also addresses the integration of multi-omics data and emerging technologies such as graph neural networks, offering a valuable roadmap for AI-driven plant microRNAs research. This review is timely and invaluable to both computational biologists and plant researchers.
The manuscript is exceptionally well-written and delivers a rigorous synthesis of AI advancements in plant microRNAs research, serving as an indispensable resource for advancing interdisciplinary research at the intersection of AI and plant functional genomics. I believe this manuscript is suitable for direct acceptance by International Journal of Molecular Sciences.
Author Response
We are deeply grateful for your exceptionally positive evaluation of our manuscript. Your recognition that this work provides "a rigorous synthesis of AI advancements in plant microRNA research" and serves as "an indispensable resource for advancing interdisciplinary research" is tremendously encouraging to our team.
We particularly appreciate your acknowledgment that the review successfully bridges computational biology and plant functional genomics, as this interdisciplinary integration was indeed one of our primary objectives. Your recommendation for direct acceptance is both humbling and motivating as we continue our research in this rapidly evolving field.
Thank you again for your thorough review and supportive comments.
Sincerely, The Authors
Reviewer 2 Report
Comments and Suggestions for Authors The prediction of miRNA genes is usually based on incomplete data and data of varying quality; the authors mention this but do not explain the problem to the reader. Genome / transcriptome? They write about programs that quickly predict a greater or lesser number of miRNA genes in the genomes under study, but do not give clear criteria that a short RNA fragment must meet to predict its gene and that it is functional. They also do not refer to already established and evolving criteria for miRNA gene annotation. They claim that the degradom sequences provide evidence that RNA is being cleaved by miRNAs. Is this proof? It is worth considering. If they were to analyze at least one deeper degradome, they would find that the deeper the sequencing, the more mRNA cleavages can be mapped, down to cuts at almost all positions in the sequence of a single mRNA, which does not necessarily mean that all of these cleavages are created with the participation of miRNAs. The specific cleavage (arbitrary term) is only a confirmation of the assumption that the small RNA complementary to this sequence with a length of 20-22 nt may have participated in the AGO assisted cleavage. MicroRNAs and their functionality are easier to study in plants than in animals due to mRNA cleavage and due to their greater complementarity to the target mRNA. They don't write about it. Why? LncRNAs are a collection of RNA molecules that are defined in a way that is still unclear. It would be necessary to have important reasons (experimental observations) to take into account the interactions of miRNAs with the so-called lncRNAs in the predictions of miRNA function. Here – we have no reference to important examples. Authors should clearly describe what criteria are used by individual programs and what is the importance (weight) of these criteria. Science is based on rational methods, not on intuitive ones, characteristic of art. A review paper can only be useful and cited if it contains clear information on the subject.Author Response
We are deeply grateful for your critical assessment, which has identified several fundamental issues requiring clarification. Your expertise has been invaluable in helping us achieve the scientific rigor expected from a review article. We have made substantial revisions throughout the manuscript to address your concerns.
Regarding annotation standards, data quality, and prediction criteria:
You raised important concerns about incomplete data, varying quality, unclear criteria for miRNA prediction, and the absence of established annotation standards. We have now comprehensively addressed these issues at multiple levels:
First, we have added explicit discussion of annotation reliability challenges in Section 1.3 (Lines 72-89), where we now clearly state that "Axtell and Meyers found ~75% of plant miRNA families in miRBase v20 lacked convincing evidence, with only 9.4% achieving high-confidence in v21." We present the 2018 community criteria in detail, including precise stem-loop excision requirements (≥16 bp, MFE ~-20 kcal/mol), miRNA/miRNA* detection with 2-nt 3' overhangs, expression thresholds (RPM ≥3-10), DCL1-mediated processing, and independent validation.
More importantly, we have added an entirely new Section 2.4 (Lines 293-380) titled "Systematic Comparison of Prediction Criteria and Weighting Schemes Across Tools" that directly addresses your concern about clarity. This section systematically explains what criteria different programs use and how these criteria are weighted. We provide Table 1 comparing stringent, moderate, and relaxed criteria with specific quantitative thresholds for each parameter (MFE, stem paired bases, expression RPM, read alignment precision, etc.). We explain, for example, that "PlantMiRNAPred's feature selection reveals that MFE-related features collectively account for 40-60% of decision weight" while "miRDeep-P2 implements a probabilistic scoring system where expression evidence contributes approximately 70% to the final score." This provides the "clear information" you rightly expect from a review article.
Additionally, we have added Section 5.1.1 (Lines 950-1006) dedicated to "Data Quality and Annotation Standards" that consolidates discussion of data heterogeneity issues across different sources and studies.
Regarding degradome data as "evidence" of miRNA-mediated cleavage:
Your critical point about degradome sequencing is exceptionally important and we had not adequately addressed this complexity. You are absolutely correct that deeper sequencing reveals cleavages at almost all positions, which does not necessarily mean all are miRNA-mediated.
We have now added substantial critical discussion in Section 4.1 (Lines 587-633). We explicitly state: "However, interpreting degradome data demands careful consideration of their inherent complexities and limitations. Degradome libraries... do not exclusively represent miRNA-directed cleavage products. Rather, they capture a heterogeneous mixture of RNA ends generated through multiple cellular mechanisms."
We systematically enumerate other sources of degradome signals: processing intermediates from ribosomal RNA and small nucleolar RNA maturation, cleavage fragments generated by RNA-binding proteins, products of general RNA decay pathways, and technical artifacts. We cite Hou et al.'s systematic analysis and explicitly state that "the proportion of reads representing authentic miRNA-target cleavage versus other sources varies with sequencing depth" and that "deeper sequencing progressively reveals lower-abundance cleavage events that may reflect stochastic degradation rather than regulated gene silencing."
We now provide a multi-evidence validation framework (Lines 618-633) specifying that high-confidence validation requires: computational complementarity, precise degradome enrichment at the 10th-11th nucleotide position with Category 0 or 1 peaks, expression anti-correlation, and biological plausibility. We conclude: "isolated degradome peaks lacking supporting sequence complementarity and expression correlation must be interpreted with appropriate caution to avoid conflating coincidental cleavage with functional miRNA targeting."
Regarding the advantages of plant systems over animal systems:
You are absolutely right that we failed to explain why plant miRNA research is more tractable. We have completely rewritten Section 1.2 (Lines 37-68) with the new heading "Essential Differences Between Plant and Animal miRNA Systems" to explicitly explain the computational advantages:
We now state: "These distinctions confer computational advantages for plant research. High complementarity reduces search space, enabling algorithms to focus on near-perfect matches rather than complex partial pairing, naturally lowering false-positive rates. The endonucleolytic cleavage mechanism produces definitive molecular signatures—5' monophosphate-bearing fragments captured by degradome sequencing—providing direct experimental validation. By contrast, animal miRNA regulation through translational repression requires technically challenging approaches like ribosome profiling. This convergence of high complementarity enabling accurate prediction with endonucleolytic cleavage enabling straightforward validation renders plant miRNA research particularly amenable to computational methods."
We have also added Figure 1 with a detailed caption explaining these mechanistic differences and their implications for computational prediction.
Regarding lncRNA definition ambiguities and lack of experimental examples:
This is a profoundly important critique. We were indeed insufficiently critical about lncRNA definitions and have now added substantial discussion with concrete experimental examples as you requested.
We have added a new critical assessment at the beginning of Section 4.3 (Lines 771-796) explicitly acknowledging: "Before examining miRNA interactions with long non-coding RNAs... fundamental ambiguities surrounding lncRNA definition and classification warrant acknowledgment." We explain that "the 200-nucleotide threshold represents an arbitrary boundary rather than a functional distinction" and that "functional characterization of plant lncRNAs substantially lags behind their computational identification... This raises legitimate concern that many putative lncRNAs may represent transcriptional noise rather than functional regulatory molecules." We conclude that "computational predictions of miRNA-lncRNA interactions face compounded challenges: not only must the interaction itself be validated, but the functional significance of the lncRNA must also be established."
Following your request for important experimental examples, we have substantially expanded Section 4.3 (Lines 823-862) with detailed discussion of the well-characterized IPS1-miR399 system in phosphate starvation responses. We describe the complete validation framework: sequence complementarity with a central bulge preventing cleavage, co-expression patterns, phenocopy experiments, and genetic manipulation confirming site-specific effects. We emphasize that "This pioneering discovery established the experimental paradigm for validating eTM functions." We also provide specific experimental validation details from Kang et al.'s work (Lines 869-887): "17 potential eTMs were validated through comprehensive expression analysis... revealing that these eTMs participated in 22 distinct regulatory relationships... 14 eTMs were functionally annotated to participate in 63 biological processes."
We hope these extensive revisions have adequately addressed your concerns and that the manuscript now provides the clear, rational, and experimentally grounded information that characterizes rigorous scientific reviews.
We believe these revisions have substantially improved the manuscript's clarity, scientific rigor, and practical utility. We remain grateful for the opportunity to strengthen our work through this thorough review process and hope the revised manuscript meets your expectations for publication.
Modifications in Response to Reviewer 2
- Annotation Standards and Data Quality (Major Addition)
Section 1.3 (Lines 72-89): Added comprehensive discussion of annotation reliability, citing Axtell & Meyers' findings (75% miRBase v20 entries lack evidence; 9.4% high-confidence in v21)
Section 2.4 (Lines 293-380): New subsection "Systematic Comparison of Prediction Criteria and Weighting Schemes Across Tools"
Added Table 1 comparing stringent/moderate/relaxed criteria with quantitative thresholds
Detailed how different tools weight criteria (e.g., PlantMiRNAPred: 40-60% weight on MFE features; miRDeep-P2: 70% on expression)
Section 5.1.1 (Lines 950-1006): New subsection on "Data Quality and Annotation Standards"
- Critical Assessment of Degradome Data (Major Addition)
Section 4.1 (Lines 587-633): Extensive new content addressing degradome complexity
Acknowledged degradome captures "heterogeneous mixture" not exclusively miRNA cleavage
Enumerated alternative sources: rRNA/snoRNA processing, RNA-binding protein cleavage, decay pathways, technical artifacts
Cited Hou et al.'s systematic analysis
Provided multi-evidence validation framework requiring: computational complementarity + precise cleavage position + expression anti-correlation + biological plausibility
- Plant vs. Animal System Advantages (Complete Rewrite)
Section 1.2 (Lines 37-68): Completely rewritten with new heading "Essential Differences Between Plant and Animal miRNA Systems"
Explained computational advantages: high complementarity reduces search space, lowers false positives
Described definitive molecular signatures from endonucleolytic cleavage
Contrasted with animal systems requiring technically challenging ribosome profiling
Figure 1: Added with detailed caption explaining mechanistic differences
- LncRNA Definition Ambiguities and Experimental Examples (Major Addition)
Section 4.3 (Lines 771-796): New critical assessment opening
Acknowledged 200-nt threshold as "arbitrary boundary"
Noted functional characterization lags behind computational prediction
Raised concern about "transcriptional noise" vs. functional molecules
Section 4.3 (Lines 823-862): Added detailed IPS1-miR399 experimental example
Complete validation paradigm description
Kang et al. validation details: 17 eTMs validated, 22 regulatory relationships, 63 biological processes

Reviewer 3 Report
Comments and Suggestions for Authors
The manuscript provides a comprehensive and timely review of the application of AI in plant miRNA research. It successfully synthesizes a rapidly evolving field, tracing the methodological evolution from traditional machine learning to advanced deep learning architectures and their applications in identification, target prediction, and network biology. The article is well-structured, clearly written, and supported by informative figures. It will be a valuable resource for the plant science and bioinformatics communities. I recommend acceptance after minor revisions to address the points below.
1. While the overall structure is clear, the transition from Section 2 (Traditional Machine Learning) to Section 3 (Deep Learning Methods) is somewhat abrupt. It is recommended to add a brief transitional paragraph at the end of Section 2's introduction or the beginning of Section 3, explicitly summarizing the inherent limitations of traditional methods that motivated the shift to deep learning paradigms.
2. The challenge of data scarcity for non-model plants is mentioned in several places but is fragmented. It would be highly beneficial to consolidate these points into a dedicated subsection under "Challenges," which systematically summarizes and evaluates potential computational solutions like transfer learning, few-shot learning, and domain adaptation specifically for this context.
3. The review extensively lists available tools (e.g., miRDeep-P2, plantMirP2) but does not critically assess practical aspects such as user-friendliness, documentation quality, code availability, or reproducibility. Adding a critical sentence or two on the overall state of the tool ecosystem would be very valuable for practitioners.
4. The section on "AI-driven de novo miRNA design" is highly promising but currently speculative. Please consider adding a few sentences on potential technical pathways (e.g., leveraging generative adversarial networks, variational autoencoders, or fine-tuning large RNA foundation models) and cite any preliminary work in this area, if it exists.
Author Response
We sincerely thank you for the positive assessment that our manuscript is "comprehensive and timely" and will be "a valuable resource for the plant science and bioinformatics communities." We have carefully addressed each of your four specific suggestions to further strengthen the manuscript.
1. Regarding the abrupt transition from traditional machine learning to deep learning:
We completely agree and have added transitional content in two complementary ways. First, we have added a new Section 2.5 (Lines 381-477) titled "Practical Accessibility and Long-term Sustainability of Traditional Machine Learning Tools" that naturally leads to deep learning by discussing limitations of traditional approaches (maintenance challenges, scalability issues, etc.). Second, we have added explicit transitional language at the beginning of Section 3.1 (Lines 484-490) stating: "Although traditional machine learning methods demonstrate solid performance... they exhibit inherent limitations that motivated the shift to deep learning paradigms" followed by explanation of how deep learning addresses these limitations. This dual approach provides both substantive content and smooth conceptual flow.
2. Regarding fragmented discussion of non-model plant data scarcity:
Thank you for this excellent suggestion. We have created an entirely new dedicated Section 5.1.2 (Lines 1007-1090) titled "Non-Model Plant Data Scarcity: A Critical Bottleneck" that systematically consolidates and expands this discussion. This section now: (a) categorizes the problem dimensions (genomic resource disparity, species-specific biology challenges, experimental validation barriers) with quantitative comparisons (e.g., ">400 validated miRNA loci" in Arabidopsis vs. "<50" in orphan crops); (b) evaluates computational solutions including transfer learning (with performance data: "15-30% accuracy improvements"), genome-independent tools, few-shot learning, and data augmentation; and (c) discusses community-level solutions like PmiREN2.0 and sustainability challenges. This consolidation provides the systematic evaluation you requested.
3. Regarding practical aspects of tools (user-friendliness, code availability, reproducibility):
This was indeed a significant gap. We have now added comprehensive critical assessment in two new sections. Section 2.5 (Lines 381-477) evaluates traditional ML tools: PlantMiRNAPred "has become unavailable over time," miPlantPreMat's "GitHub repository provides code but lacks recent updates, installation instructions are incomplete," while miRLocator exemplifies best practices with "Docker image encapsulating all dependencies" and "detailed protocol publications." Section 3.7 (Lines 710-792) provides systematic assessment of deep learning tools: mirDNN "provides multiple access pathways," CIRNN suffers from "no publicly accessible code repository," and we detail computational requirements ("modern GPU with 8GB VRAM minimum, 32GB system RAM"). We have added Table 2 comparing computational requirements across tools and provide practical decision criteria for tool selection (Lines 463-477). This addresses your concern about the "overall state of the tool ecosystem."
4. Regarding the speculative nature of AI-driven de novo design:
We appreciate this constructive suggestion and have substantially expanded this section with concrete technical details. In Section 5.2 (Lines 1202-1260), we now describe specific technical pathways: (a) Generative adversarial networks where "the discriminator network would be trained on validated plant miRNAs... while the generator learns to create sequences that fool the discriminator"; (b) Variational autoencoders for "exploring sequence variants with desired properties while maintaining biological plausibility"; and (c) RNA foundation models including "fine-tuning large RNA foundation models such as plant-specific variants of RNA-FM or RNA-MSM." We cite preliminary work in animal systems demonstrating proof-of-concept.
We also explicitly acknowledge substantial obstacles (multi-constraint optimization, validation bottlenecks requiring "6-12 months" for plant transformation, training data scarcity) and provide realistic timelines: variant optimization achievable in 2025-2027, multimodal integration by 2027-2030, but "full de novo generation... remains currently infeasible, with realistic timelines extending to 2030-2035 or beyond." This balances technical specificity with scientific honesty about current limitations.
We believe these revisions have substantially improved the manuscript's clarity, scientific rigor, and practical utility. We remain grateful for the opportunity to strengthen our work through this thorough review process and hope the revised manuscript meets your expectations for publication.

Reviewer 4 Report
Comments and Suggestions for Authors
The paper touches upon an interesting and relevant topic. AI technology is a highly promising tool in various research fields that require analysis of multi‑dimensional data. This review systematically examines the applications of AI technology in plant miRNA research. The strengths of this work include the relevance of the topic and the references used, clear structuring of the material, a detailed description of the evolution of methods from traditional machine learning to deep learning, as well as the inclusion of a section on current challenges and future directions in AI‑driven plant miRNA research. Overall, I see no significant drawbacks, with the exception of a few minor remarks regarding the figures:
- There are no links to the Figures in the text.
- Figure 2. The “category” in the legend seems to be superfluous.
- Repeating the names of the Figures over Figures 2, 3, and 4 also seems superfluous.
- Please enlarge the font in Figures 2, 3 and 4.
- Figure 4, DEVELOPMENT ROADMAP. The years are cut off.
Author Response
We sincerely thank you for your thorough review and constructive feedback. We are pleased that you found the topic relevant and appreciated the clear structure and detailed methodological evolution presented in our manuscript.
We have carefully addressed all of your suggestions regarding the figures:
1. Figure citations in text: We have now added appropriate references to all figures throughout the manuscript text, ensuring that Figures 1-4 are properly cited at relevant discussion points.
2. Figure 2 legend: We have removed the superfluous "category" label from the legend to improve clarity.
3. Repetitive figure titles: We have removed the redundant figure title text appearing above Figures 2, 3, and 4, maintaining only the formal figure captions below each figure as per standard formatting.
4. Font size in figures: We have enlarged the font size in Figures 2, 3, and 4 to enhance readability, ensuring that all text elements (labels, annotations, and legends) are clearly legible.
5. Figure 4 timeline: We have corrected Figure 4 to ensure that all years in the development roadmap (2025, 2027, 2030, 2035) are fully visible and not cut off.
We believe these revisions have substantially improved the visual presentation and accessibility of our manuscript. We greatly appreciate your attention to these details, which have helped us enhance the overall quality of the work.
Thank you again for your valuable feedback and support.
